computer modelling and simulation/artificial intelligence/graph theory

game theory, Prisoner's Dilemma, cooperation, graph topology

**Author for correspondence:**
C. O'Riordan
e-mail: colm.oriordan@nuigalway.ie

# Introducing a graph topology for robust cooperation

## A. M. Locodi and C. O'Riordan

National University of Ireland Galway, Computer Science, Galway, Ireland

CO, 0000-0003-0449-8224

Identifying the conditions that support cooperation in spatial evolutionary game theory has been the focus of a large body of work. In this paper, the classical Prisoner's Dilemma is adopted as an interaction model; agents are placed on graphs and their interactions are constrained by a graph topology. A simple strategy update mechanism is used where agents copy the best performing strategy of their neighbourhood (including themselves). In this paper, we begin with a fully cooperative population and explore the robustness of the population to the introduction of defectors. We introduce a graph structure that has the property that the initial fully cooperative population is robust to any one perturbation (a change of any cooperator to a defector). We present a proof of this property and specify the necessary constraints on the graph. Furthermore, given the standard game payoffs, we calculate the smallest graph which possesses this property. We present an approach for increasing the size of the graph and we show empirically that this extended graph is robust to an increasing percentage of perturbations. We define a new class of graphs for the purpose of future work.

## 1. Introduction

Understanding the emergence and robustness of cooperation in environments where agents or individuals are tempted to behave in a non-cooperative manner has been the focus of many studies. The Prisoner's Dilemma is a frequently adopted game to help explore these questions.

In this game, an agent can either cooperate (C) or defect (D), obtaining a payoff that depends on both agents' choices [1]. The payoffs are as follows: T (temptation to defect, i.e. the payoff for defection when the other player cooperates), R (reward for mutual cooperation), P (punishment for mutual defection) and S (the sucker's payoff for cooperation when their opponent defects). The constraint that $T > R > P > S$ holds, the payoffs can be seen in the payoff matrix in figure 1. Many variants of the Prisoner's Dilemma have been explored including *N*-player variants [2,3], games with optionality [4,5] and games with non-discrete payoffs allowing partial cooperation and partial defection [6,7].

|  | player 1 | |
|---|---|---|
|  | cooperate | defect |
| cooperate | (*R*,*R*) | (*S*,*T*) |
| defect | (*T*,*S*) | (*P*,*P*) |

(player 2 is the left-side label for the rows)

**Figure 1.** Payoff matrix.

Spatial game theory involves constraining interactions between agents by adopting some topology which governs which agents can interact together [8]. Agents are placed on nodes in the graph and an edge connecting two nodes indicates that the agents on these nodes can interact with each other [9].

Research over the past years has shown that different topologies governing agent interactions such as lattices [10], scale-free graphs [11–13], small-world graphs [14,15], graphs exhibiting community structure [16] and bipartite graphs [17,18] have a considerable impact on the emergence of cooperation. In much of the research, agents are randomly assigned a strategy and the focus of the research is to explore whether cooperation or defection emerges. An alternative methodology that has gained considerable attention is to seed the population with cooperators and then introduce a defector (or defectors) and analyse the effect on the population. This approach was explored in the seminal paper by Nowak & May [19]. This has parallels with research in the domain of percolation theory [20].

Our game uses synchronous updates, all the players calculate their payoffs and potentially update their strategy at the same time. A player's payoff is calculated as the sum of all payoffs obtained by the player when interacting with their neighbours in the current turn. Players update their strategy by copying the strategy of their best-performing neighbour. This approach is one of several approaches that have been adopted in the literature; others include copying a better performing strategy with a probability proportional to their score and the Fermi–Dirac strategy update approach [21]. Before the start of a simulation we seed the population by setting all the nodes to cooperators, then turning a percentage of them to defectors at random.

The simulation is done using the algorithm specified in Algorithm 1:

In our simulations, we set 'max time-step' to be 1000; however, it was never reached as every simulation converged before then. The simulator can be found on GitHub [22].

Our work has a strong relation to the notion of evolutionary amplifiers and suppressors [23]; given an increased payoff due to an advantageous mutation in one of the players in a population the fixation probability of the mutant (the probability that the mutation will spread to the whole population) can be increased in some graphs called amplifiers and decreased in other graphs called suppressors.

There has been extensive work undertaken investigating the effect of topology on the spread of behaviours in spatially organized populations. Early work focused on lattices with more complex graph exhibiting a much wider range of features. The topology described in this paper has some key similarities with existing work—the presence of star-like graphs which have been investigated on their own and which are also a component of comet graph structures [24].

The effect of the topology on the outcome is dependent on the interaction model; much of the recent work adopts an abstract interaction model where mutants who outperform the initial population are considered. In our work, we consider a more complex interaction model (the Prisoner's Dilemma) where the payoff received by an agent is dependent on its neighbours. The increased complexity of this interactions model requires a more complex topology to help maintain robust cooperation.

**Algorithm 1:** Simulation algorithm.

current time-step = 0

**while** *current time-step < max time-step* **do**

 **if** *this is not the first time-step* **then**

 **for** *each node* **do**

 update its current strategy to its next strategy;

 **end**

 **end**

 **for** *each node* **do**

 calculate its payoff by playing a game with each of its neighbours;

 **end**

 **for** *each node* **do**

 determine its next strategy by identifying the strategy of its best performing neighbour, if its payoff is higher then the payoff of the current node, in case of ties pick at random;

 **end**

 **if** *All the nodes did not change strategy in the last 10 time-steps* **then**

 end simulation

 **end**

 current time-step ++

**end**

Subgraphs of our topology, critical node subgraphs, have similar functionality to that of 'self reinforcing' structures [25]; they both stop other strategies/conventions from outside the subgraph to spread into the subgraph, but our subgraph differs in the sense that they are also responsible for spreading the cooperation to nodes outside the subgraph, nodes which in their turn spread cooperation further due to the subgraph they belong to; we present proof of this property. Another key feature of the topology presented in this work is that the robustness to perturbations (the introduction of defectors) increases as the graph size increases. While the graph topology shares some ideas with existing approaches, one of the novel aspects of this topology is that it operates with the Prisoner's Dilemma as the interaction model.

The main contributions of this work are: firstly, the introduction of a graph topology that ensures a cooperative population is robust to any one perturbation, and secondly, a means to increase the size of the graph such that it is robust to all single node perturbations and depending on the size of the graph, robust to many node perturbations. We show that the graph can be increased in size indefinitely rendering it more robust as it is extended. We compare the robustness on our graph topology with that of the star graph.

The paper outline is as follows: in §2, we present the graph topology and a sketch for the proof, with the full proof, and associated derived constraints included in appendix A. There are further discussions in appendices B, C and D. In §3, a description of a method to extend the graph while maintaining its functionality is presented. In §4, we empirically test the robustness in graphs of increasing size and show that robustness improves as the graph extends. In §5, we identify by empirical means the payoff vales for which our graph can exist. In §5, we also define a type of graph named 'Resilient Graphs' (with our graph belonging to this type), compare them to evolutionary suppressors, and show a resilient graph for defectors; we then discuss the effects of several other update mechanisms on our topology, the star graph and the complete graph, followed by how our topology might be useful for finding resilient graphs in the optional Prisoner's Dilemma interaction model, how sufficiently large random, scale-free or real-world graphs might be modified to contain our topology; we discuss robustness in our topology and other graphs, and we look at the effects on our topology of a more targeted initial perturbation. Finally, in §6, we present conclusions and outline some future research directions.

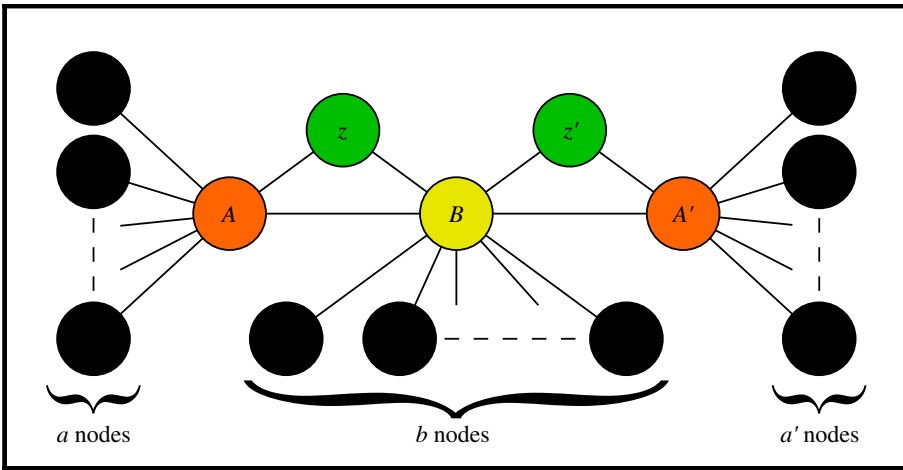

**Figure 2.** Locodi graph.

## 2. Graph topology

The graph topology presented herein possesses some interesting properties. The general topology has some properties present in many real-world graphs, i.e. the existence of hubs and the existence of many nodes with low degree. The graph is useful for guaranteeing robustness to any single node perturbation (i.e. a cooperator changing to a defector). The basic topology is depicted in figure 2.

With reference to the graph depicted in figure 2, we can identify four different classes of nodes (colour coded in the diagram):

— The $a$, $a'$ and $b$ nodes are minor nodes that have a degree of one. The number of $a$ nodes, $a'$ nodes and $b$ nodes present is important in determining if the graph is robust to perturbations; their numbers need to satisfy certain requirements to ensure the graph supports a population's robustness to any one perturbation. The graph does not need to be symmetric, the number of $a$ nodes can be different to the number of $a'$ nodes.

— The $A$ and $A'$ nodes are referred to as critical nodes. If, at any stage, a critical node and all its minor nodes are cooperators, then cooperation will spread to all the graph.

— The two critical nodes are connected via the $B$ node. The constraints on this node guarantee that if it defects, it cannot cause defection to spread to the critical nodes. Moreover, when the $B$ node and its minor nodes cooperate, the $B$ node can cause a critical node that has defected (and all its minor nodes) to cooperate again.

— The $z$ and $z'$ nodes are pivotal in allowing cooperation to spread between two large nodes (those nodes with high degree i.e. $A$, $A'$ and $B$) and their minor nodes; we refer to them as enabler nodes.

Given the local topology of these different classes of nodes, these nodes behave differently when a node is perturbed. For each class of node identified, the graph will revert to full cooperation. We quickly enumerate these cases below and include a full proof in appendix A.

— If one of the critical nodes ($A$ or $A'$) defects, as seen in figure 3:
  Time-step 1: All the minor nodes and the enabler node connected to the initial defector will defect. If the payoff of the other critical node is lower than the initial defector, node $B$ will defect and we continue to Time-step 2 otherwise we jump to Time-step 4.
  Time-step 2: The minor $b$ nodes will defect; the other critical node will ensure the $B$ node cooperates again.
  Time-step 3: The minor $b$ nodes will cooperate due to the presence of the $B$ node.
  Time-step 4: The initial defector and the enabler node connected to it will cooperate due to the presence of the cooperative $B$ node.
  Time-step 5: All the minor $a$ or $a'$ nodes will cooperate again given their connection to the initial defector.
— If the $B$ node defects:
  Time-step 1: All the minor $b$ nodes will defect due to the presence of the $B$ node; the $B$ node will cooperate due to the presence of the critical nodes.
  Time-step 2: All the minor $b$ nodes will cooperate due to the presence of the cooperative $B$ node.

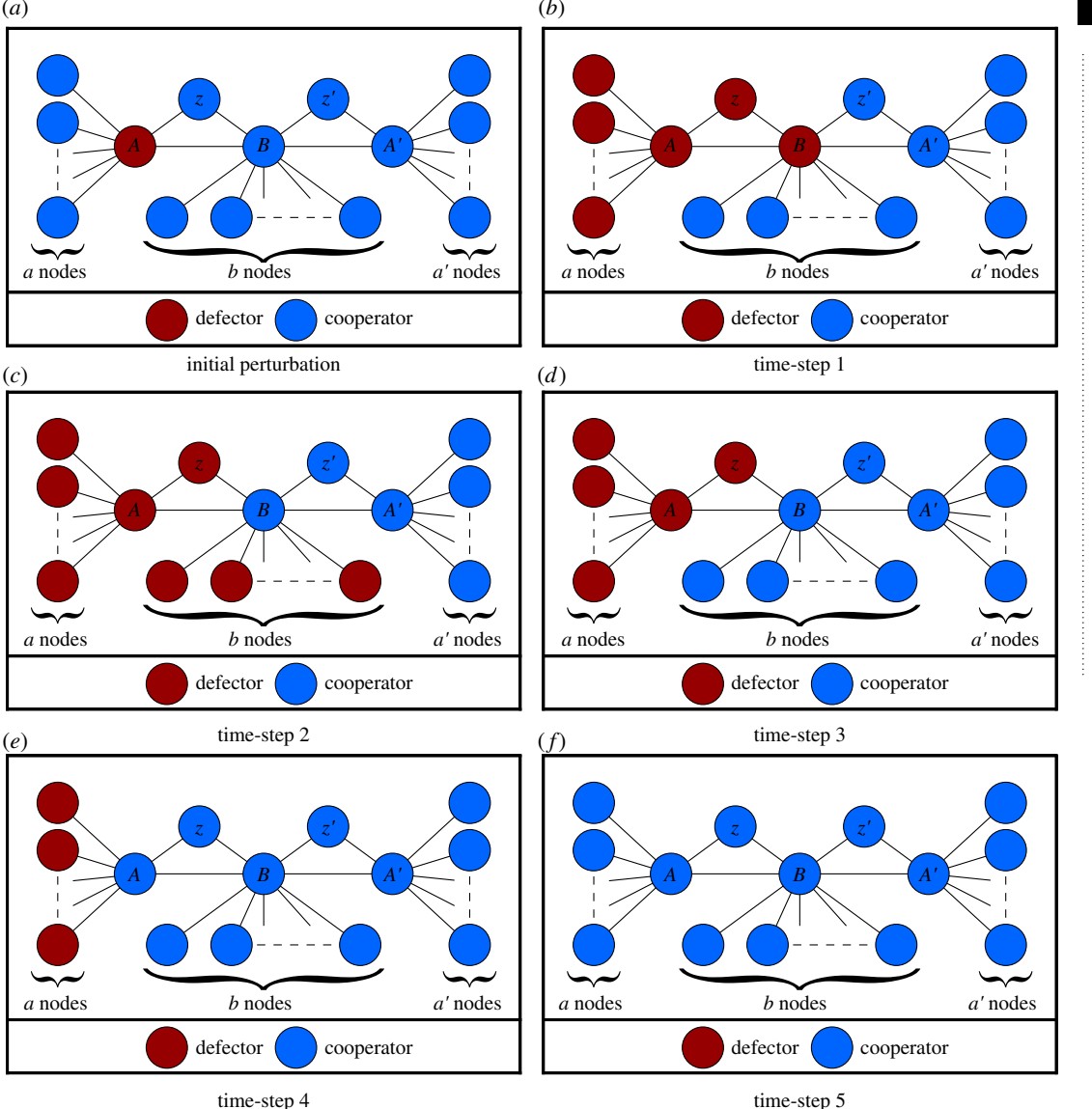

**Figure 3.** Simulation steps when node *A* defects. (*a*) Initial perturbation, (*b*) time-step 1, (*c*) time-step 2, (*d*) time-step 3, (*e*) time-step 4, (*f*) time-step 5.

— If one of the *a*, *a'*, *b*, *z* or *z'* nodes defects, it will revert to cooperation in the following time-step due to their connection to a cooperative large node.

A thorough proof with illustrations is presented in appendix A together with a specification of all the constraints needed on the graph topology. In appendix B, we present additional requirements which are needed to allow a critical node to continue to cooperate if it, and all of its minor nodes, cooperate. The constraints on the graph topology are further examined in appendix C. In appendix D, we show that the smallest such graph structure adopting the classical Prisoner's Dilemma payoffs, ($T = 5$, $R = 3$, $P = 1$ and $S = 0$), has $|a| = |a'| = 16$ and $|b| = 6$.

Below are the list of requirements obtained in appendices A and B:

(1) $2T < R(a + 1) + S$
(2) $2T < R(a'+1)+S$
(3) $2T < R(b + 3) + S$
(4) $T(b + 4) < R(a + 1) + S$
(5) $T(b + 4) < R(a'+1)+S$
(6) $S \times b + 4R > T$
(7) $R(a' + 1)+S>P(a + 2)$
(8) $R(a + 1) + S > P(a'+2)$

(9) $S(b + 2) + 2R > T$

(10) $P(a + 1) + T < R(a'+2)$

(11) $P(a'+1)+T < R(a+2)$

(12) $P(a + 1) + T < R(b + 2) + 2S$

(13) $P(a' + 1) + T < R(b + 2)+2S$

(14) $T < S \times a + 2R$

(15) $T < S \times a'+2R$

(16) $T + P < R \times a + 2S$

(17) $T + P < R \times a'+2S$

(18) $T(b + 3) + P < R \times a + 2S$

(19) $T(b + 3) + P < R \times a'+2S.$

# 3. Extending the graph

We can extend the graph by adding additional nodes and edges to the graph and can possibly modify a graph by removing existing ones. We may wish to consider certain properties of the graph when we decide upon a mechanism to extend the graph.

One relatively straightforward mechanism to extend the graph is by creating additional critical nodes ($A$) and large nodes ($B$) and connecting them together in a 'line' topology as depicted in figure 4. Note that the minor nodes and the enabler nodes are omitted from the figure as the overall 'shape' of the graph is determined by the large nodes. With this 'line' topology we can increase the size of the graph indefinitely and we adopt this approach in the next section to show the effects of extending the graph on the robustness of the graph. Given the basic graph structure, the graph is guaranteed to be robust up to any $n - 1$ perturbations where $n$ is the number of critical nodes present in the graph.

Given the definition of our graph, the number and ratios of minor nodes can be varied while maintaining the core functionality of the graph; we have identified a series of additional methods we can use to grow or extend our graph while maintaining the core functionality. These include:

1. We can add additional enabler nodes: this may alter the degree distribution of our graph potentially allowing for other payoffs and could make cooperation spread faster in some cases.
2. We can add edges between minor nodes both connected to the same large node or to different large nodes. This may alter the degree distribution of our structure. It could slow the spread of cooperation. There is a limit to which we can do this before losing the core functionality.
3. A critical node can be connected to multiple large non-critical nodes ($B$) (this will need the addition of enabler nodes).
4. A large non-critical node can be connected to other large non-critical nodes (this will need the addition of enabler nodes).
5. A large non-critical node can be connected to multiple critical nodes (this will need the addition of enabler nodes).

Given points 3 and 5 above, if we consider a critical node subgraph to be a 'node' and the large non-critical node subgraph connecting two critical node subgraphs to be the 'edge' between two 'nodes', then one can create any undirected 'graph' mimicking some of the properties of those actual graph. Examples include fractal graphs and scale-free graphs.

The many ways to grow our topology and their effects on the graph properties and the limit to which we can maintain core functionality will be explored in future work.

# 4. Robustness

To demonstrate the effects of extending the graph size on the robustness of cooperation to perturbations, we look at four different graphs, all of different size but adopting the same 'line' topology. The details of the graph sizes are specified in table 1.

We test the robustness of the graph to perturbations by beginning with a fully cooperative graph and randomly perturbing some nodes (changing a cooperator to a defector). We vary the percentage of nodes to be perturbed and measure the ability of the population to recover to a fully cooperative state.

To demonstrate the effects of extending the graph on the robustness of the graph, we initially look at graphs with 88 nodes of which 3 are critical nodes, then we extend the graph three times: the first one has

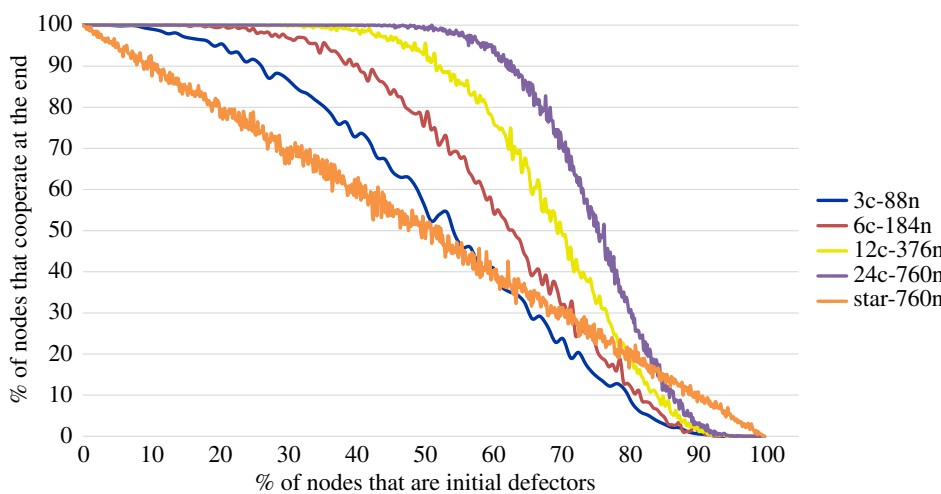

**Figure 4.** Line topology.

**Figure 5.** Robustness of graphs to perturbations.

**Table 1.** Details of the graph sizes.

| no. of nodes | no. of critical nodes | no. of $B$ nodes | no. of enabler nodes | no. of $b$ nodes | no. of end $a$ nodes | no. of middle $a$ nodes |
|---|---|---|---|---|---|---|
| 88 | 3 | 2 | 4 | $2 \times 9$ | $2 \times 21$ | $1 \times 19$ |
| 184 | 6 | 5 | 10 | $5 \times 9$ | $2 \times 21$ | $4 \times 19$ |
| 376 | 12 | 11 | 22 | $11 \times 9$ | $2 \times 21$ | $10 \times 19$ |
| 760 | 24 | 23 | 46 | $23 \times 9$ | $2 \times 21$ | $22 \times 19$ |

184 nodes of which 6 are critical nodes, the second one has 376 nodes of which 12 are critical nodes and the last one has 760 nodes of which 24 are critical nodes. We test the robustness, using classical Prisoner's Dilemma payoffs, for all the four graphs; and as seen in figure 5, increasing the graph significantly increases the robustness of the graph. In the last graph, robustness to perturbations is very high even at 50% perturbations. For all the graphs, we use the line topology.[1]

In the figure, we also present the robustness for a star graph, a graph which has one node connected to all the other nodes and all the other nodes having a degree of 1, with 760 nodes. The star graph has an average percentage of cooperators at the end of simulation of approximately $100 - x$, where $x$ is the percentage of initial defectors. As we can see, the star graph is outperformed by our topology up to a certain perturbation threshold which increases as the size of our topology increases.

From these results, we can determine that populations become more robust to the introduction of defectors as the graph extends; in particular, the robustness increases as the number of critical nodes increases. The data can be found on GitHub [22].

# 5. Discussion

For the interaction model and update mechanism adopted in this work, the actual values of the payoffs in the payoff matrix are not important; what is important is the ratios of the payoffs. For example, for the

---

[1]Due to the manner in which the original graph is defined, the result of a simulation is either that all the nodes cooperate or all of them defect, so the results in the figure tell us how likely it is for the graph to fully cooperate given that a certain percentage of the nodes initially perturbed from cooperation to defection.

two following sets of payoffs: $T = 10$, $R = 5$, $P = 2$, $S = 1$ and $T = 100$, $R = 50$, $P = 20$, $S = 10$, the simulations will behave in exactly the same way. It is not the actual values but the ratios of the payoff that are of importance for the simulations; for instance, in the example above, the ratio $T/R = 2$ is one of the ratios that will guide the simulation.

In order to identify for which payoff combinations our graph can exist, an empirical investigation is undertaken whereby simulations are run for a large set of payoff ratio values. We have looked at three payoff ratio variables $T/R$, $R/P$ and $P/S$ and we have explored all payoff ratio values from 1.1 to 9.9 with increments of 0.1. For each set of values, we attempt to find the smallest values for $a = a'$ and $b$, the number of minor nodes, for which we can respect all the rules listed in appendix C. We searched up to a maximum value of 20 000 (the largest minimum value of $a$ used in a solution was $a = 12\,445$). The simulations ended up finding a solution to our graph as long as $T/R < R/P$ (this requirement has also been obtained in appendix C).

We have performed similar simulations for the special case of $S = 0$ for which we only have two payoff ratio variables $T/R$ and $R/P$. Again the simulations ended up finding a solution to our graph as long as $T/R < R/P$, but given $S = 0$ the requirements for our graph also need $T/R < 2$. Note, we can increase the value of the ratio by adding more enabler nodes.

We introduce the following category of graphs, 'Resilient Graphs', a resilient graph is a graph which will support one strategy while hindering all others. Resilient graphs are a more generalized version of suppressors, which make mutants less likely to spread to the whole population. For Resilient Graphs, there is no need to know if a strategy/mutation is advantageous which allows us to look at more complex games. The graph topology presented in this paper is a resilient graph for cooperators participating in the Prisoner's Dilemma with a strategy update mechanism where the best performing player in the neighbourhood is copied.

A simple example for a resilient graph for defectors in the Prisoner's Dilemma interaction model and update mechanism involving copying the best performing player in the neighbourhood is the complete graph. Given a population of $n$ players all of which are defectors and perturbing $x$ (with $x < n$) of them to become cooperators, then the payoff for any cooperator is $R(x - 1) + S(n - x - 1) + S$ while the payoff for any defector is $T(x - 1) + P(n - x - 1) + T$, the payoff of cooperators is smaller than that of defectors, making it so all cooperators will revert to being defectors.

While the fixation probability represents the chance that the mutant will spread to the whole population, our robustness measurement, the average percentage of cooperators at the end of simulation given a certain percentage of initial defectors, also takes into consideration outcomes in which the mutation only spreads to a subset of the population, with the graph reaching a stable state in which both the original population type and the mutants exist at the same time. By looking at a percentage of nodes where the strategy is perturbed/mutated we can account for the effects the size of a graph may have on the robustness scores. We can better compare graphs by looking at a variety of perturbation/mutation percentages. It is possible some graphs will perform better when there is a high amount of initial perturbations.

While the structure of a population affects the overall behaviour of that population, the actual effect of the structure is highly dependent on the interaction model and update mechanism used. We will discuss a few of the many possible interaction models and update mechanisms and their effects on our structure, the star graph and the complete graph.

In the update mechanism presented in this paper all nodes update their strategy at the same time (synchronous update). With a different update mechanism which uses asynchronous updates, where only one node updates at a time, the behaviour of the population might change. We will discuss two types of asynchronous updates:

— When a node updates, all nodes recalculate their payoff: Our topology is still robust to any one perturbation. Eventually, defection spreads enough such that cooperation can spread back. Defection still cannot spread from node $B$ to node $A$ or from node $a$ to node $A$ or from node $z$ to either node $A$ or $B$. There may be some changes to the overall robustness of the graph, and the graph will take many more turns to stabilize (either to full cooperation or defection). The star topology and the complete graph have the same robustness, but again it will take multiple turns for the population to stabilize.

— A node recalculates its payoff only when it updates (once initially before comparing with others and once the update finishes): Our topology is still robust to any one perturbation. As in the previous case, defection eventually spreads enough so that cooperation can spread back. As before, defection still cannot spread from node $B$ to node $A$ or from node $a$ to node $A$ or from node $z$ to

either node $A$ or $B$. A defective critical node should maintain a higher payoff for longer (we have to wait for its minor nodes to defect and then it has to update); eventually cooperation will spread once the defected critical node updates to a lower payoff. Again there might be some changes on the overall robustness of the graph and the graph will take many more turns to stabilize. The star topology will obtain the same robustness. For the complete graph robustness stays the same, it is possible for cooperation to spread (for example, after initial perturbation one of cooperators is selected then it will defect, and then if all other defectors are selected one at a time then no defector will have the original payoff which when the next defector is selected may allow the cooperators, who have their original payoff, to have a higher payoff allowing cooperation to spread) but at no point can cooperation spread to more than the original number of cooperators, which means defection will eventually spread to the whole graph.

Given our current interaction model and update mechanism, the payoff of a node is highly dependent on the degree of that node. Generally, as a node's degree increases so does their payoff. It would be interesting to explore an alternative payoff mechanism where we normalize the payoff, by dividing it by the node's degree, resulting in an average interaction payoff. Given this approach, the topology presented in this paper will no longer function in the same manner. For example, if a minor node defects it will always have a payoff of $T$ when the large node it is connected to cooperates. The maximum payoff of this large node is $R$, which means cooperation cannot spread to the perturbed minor node and in fact it is guaranteed to spread defection. This implies that star graphs or structures containing them are not suitable for maintaining cooperation for this particular interaction model and update mechanism. In the complete graph if we have $m$ defectors and $n$ cooperators the payoff of defectors is $(n \times T + m \times P - P)/(n + m)$ and that of cooperators is $(n \times R + m \times S - R)/(n + m)$ given $T > R > P > S$ then the defectors will always have a higher payoff than cooperators.

In the current model, the players have perfect knowledge and behaviour. In an update mechanism that contains noise, this is no longer the case. We discuss two types of noise:

— Noise giving the player a chance to incorrectly read the payoff obtained by a neighbouring node when it decides how to update: We can no longer guarantee that our structure is fully robust to any one perturbation (for example if $B$ defects, both $A$ and $A'$ might read its payoff as being larger than theirs, which would allow the defection to spread to them, which eventually would probably make the defection to spread to all nodes). The overall robustness of the graph would also be affected the extent of which is hard to say since the noise will help with the spread of both defection and cooperation. To reduce the effects of the noise on our structure, we could have more minor nodes and enabler nodes. The change in payoff caused by noise will need to be very high to affect the robustness of the star topology (and even higher for larger stars). In a complete graph, it will now be possible for a defector to read a cooperator's payoff as being higher, which will spread cooperation to the defector, at the same time defection will probably spread to the cooperator (and most of all the other cooperators), so on average the robustness of the complete graph should stay the same.

— Noise giving the player a chance to incorrectly read the strategy of a node (with no change to its payoff): We can no longer guarantee that our structure is fully robust to any one perturbation. In a fully cooperating critical node subgraph, the critical node still cannot be turned to defection; the $B$ nodes will read $A$ nodes as defectors from time to time causing them to defect, minor nodes will turn to the opposite strategy of the critical nodes due to noise which makes the critical node weaker when it cooperates possibly allowing the $B$ node to influence it, it also makes defector critical nodes stronger. It is hard to predict how much this noise will reduce the overall robustness of our structure, but is expected that at all times a fraction of the minor nodes will always be defective due to wrongly perceiving the strategy of their cooperating large nodes. To combat the effect of this noise, we can increase the number of $a$ and $b$ nodes such that the ratio $a/b$ becomes larger making it more difficult for node $B$ to influence node $A$. On a star graph, the overall robustness of the graph will still mostly depend on whether or not the central node is the initial perturbation, but the robustness will further be reduced since at all times a fraction of the one-degree nodes will defect due to the noise. On the complete graph, a fraction of the nodes will cooperate at all times due to noise, but this is reduced since the perceived cooperator will have the same payoff as the defectors so the selected strategy is randomly selected from one of the defectors or the defectors incorrectly read as cooperators, increasing the overall robustness for cooperation.

As we have discussed above, our structure will still maintain some, or all, of its functionality with some other update mechanisms. Similarly, the structure could perform the same role of promoting cooperation for some other interaction models, or it could be modified slightly or be incorporated as a subgraph of a structure that has that role. For example, we can consider a different interaction model such as the Optional Prisoner's Dilemma in which the players have a third option, to abstain, which gives both players a payoff of $L$ with $R > L > S$. In this case, our structure will still work when there are only cooperators and defectors but when we introduce abstainers, it is possible that cooperation would not spread due to the presence of abstainers (for example from a cooperating $B$ node subgraph to an abstaining $A$ node subgraph when $L > P$), it is possible that a Resilient graph for cooperation for the Optional Prisoner's Dilemma can be derived by modifying or incorporating our structure.

When looking at sufficiently large random, scale-free or real-world graphs, we would generally find star-like subgraphs in them. Given this, it would be possible to slightly modify graphs such that our graph is present as a subgraph, which would improve the robustness of the whole graph due to the robustness of our subgraph.

One of the reasons it is hard to compare the robustness of two graphs is that this robustness is not only dependent on the interaction model and update mechanism but also on the actual payoffs used. As we have discussed, our structure only functions properly for a certain range of payoffs, whereas, for example, the robustness of a $k = 4$ lattice graph (a graph in which each node is placed on a grid and it is connected to its four closest neighbours with the nodes on the edges being neighbours with the nodes on the opposite edge), will vary considerably across multiple ranges of payoffs (the behaviour will be different when a defector connected to two cooperators and two defectors has either a higher payoff then a cooperator connected to three cooperators and a defector compared with the case in which it is lower). In general, for sufficiently large graphs our topology should outperform others, for the payoffs in which it functions properly, given the fact that its robustness improves as it grows while that of others should in general stay unchanged.

In our experiments, the initial perturbations are at random. If instead the perturbations were more targeted with higher degree nodes selected more often, then it will lower the robustness of our structure. The exact effects would depend on the strength/precision of targeting; for example, if half the critical nodes are guaranteed to be targeted while all other perturbations are at random, then the robustness of the graph would be very similar to that of random targeting in a graph of half the size. If critical nodes are 20 times more likely to be targeted compared with other nodes, then the robustness of the graph would be very similar to that of random targeting in a graph 20 times smaller. As discussed in the definition of critical nodes, if at any time a critical node and all its minor nodes cooperate, then cooperation will spread from it. Regarding our structure, as seen in figure 2, we showed that it is robust to any one perturbation; when considering two perturbations the graph recovers to full cooperation as long as one of the critical nodes is not perturbed (given the definition of critical nodes and the fact that node $B$ would have a lower payoff then the defected critical nodes). To reduce the effects of more targeted initial perturbations, our structure would be constructed such that the large nodes would have the minimum number of minor nodes, reducing the chance that the large node is targeted; we can also add additional edges between minor nodes making them more likely to be targeted. This targeted perturbation will similarly reduce the robustness of the star graph, while for the complete graph it will have no effect since all nodes in it have the same degree.

# 6. Conclusion and future work

In this paper, we presented a topology which supports robust cooperation in spatial evolutionary game theory; the classical Prisoner's Dilemma is adopted as an interaction model; agents are placed on graphs and interact with their neighbours. A simple strategy update mechanism is used where agents adopt the best performing strategy of their neighbourhood (including themselves). We showed that the graph is robust to one perturbation (§2), calculated the smallest such graph (appendix B) using the standard payoffs, and showed empirically that the graph is more robust as we extend it (§4).

Future work will examine the ways of extending the presented graph topology and how it affects the graph properties. We will also look at finding resilient graphs for the optional Prisoner's Dilemma in which players can choose to abstain, for weighted graphs, for normalized payoff update mechanism, for other interaction models and other strategy update mechanisms. We will explore the

robustness of standard graphs when using the same interaction model and update mechanism as in this paper. Furthermore, for the presented interaction model and update mechanism, we will show graphs in which the spread of cooperation and defection behaves in a manner that mimics logic gates in circuits, these graphs share many topological similarities with the graph presented in this paper. Once Resilient structures have been explored in many interaction models and update mechanisms, we can take a look at what common characteristics these structures have and see if some characteristics are only viable for a subsection of interaction models and update mechanisms.

Data accessibility. Source code for the simulator and related data in the paper is available at: https://doi.org/10.5281/zenodo.4562672.
Authors' contributions. The first author is responsible for all code, for the design of the topology, proof and writing of sections of the paper. The second author contributed to the writing of the paper, the experimental design and advice on the proof in the paper.
Competing interests. We declare we have no competing interests.
Funding. This work was supported by the National University of Ireland Galway College of Engineering and Informatics Postgraduate Research Scholarship.

# Appendix A

## A.1. Graph structure

Notation:

— $P_x$ is the payoff of a node $x$.
— $T$, $R$, $S$ and $P$ refer to the classical payoffs in the classical Prisoner's Dilemma, i.e. $T > R > P > S$; The payoffs are positive and real.

*Proof.* We can consider several different cases for the single perturbation, i.e. different types of nodes can be perturbed. In other words, the agent located on that node changes from cooperation to defection:

1. node $z$ or $z'$
2. one of the $a$ or $a'$ nodes
3. one of the $b$ nodes
4. node $B$
5. node $A$ or $A'$

Initially, all the nodes are cooperative, as depicted in figure 6 below.

For each case, the figures show how the graph will look like at the end of a time-step, after the update.

**Case 1: One of the $z$ or $z'$ nodes is perturbed**

At time-step $t_0$, the initial node changes to defection (figure 7).

At time-step $t_1$, the payoff of $A$ (or $A'$) and $B$ is bigger than that of the defector so the initial defector, $i$, will copy their strategy and begin to cooperate.

At that moment, we have the following payoffs:

$$P_B = R \times b + R \text{ (due to } A) + R \text{ (due to } A') + R \text{ (due to } z') + S \text{ (due to } z) = R(b + 3) + S$$
$$P_z = T \text{ (due to } A) + T \text{ (due to } B) = 2T$$

Similarly for $P_{z'}$

$$P_A = R \times a + R \text{ (due to } B) + S \text{ (due to } z) = R(a + 1) + S$$

Similarly for $P_{A'}$

This results in the following requirements:

$$P_z < P_A, \ P_{z'} < P_{A'} \text{ and } P_z < P_B \Leftrightarrow$$

$$\mathbf{2T < R(a + 1) + S} \tag{A 1}$$
$$\mathbf{2T < R(a' + 1) + S} \tag{A 2}$$
$$\mathbf{2T < R(b + 3) + S.} \tag{A 3}$$

**Case 2: One of the $a$ or $a'$ nodes is perturbed**

At time-step $t_0$, the initial node defects (figure 8). Let $i$ be the initial defector.

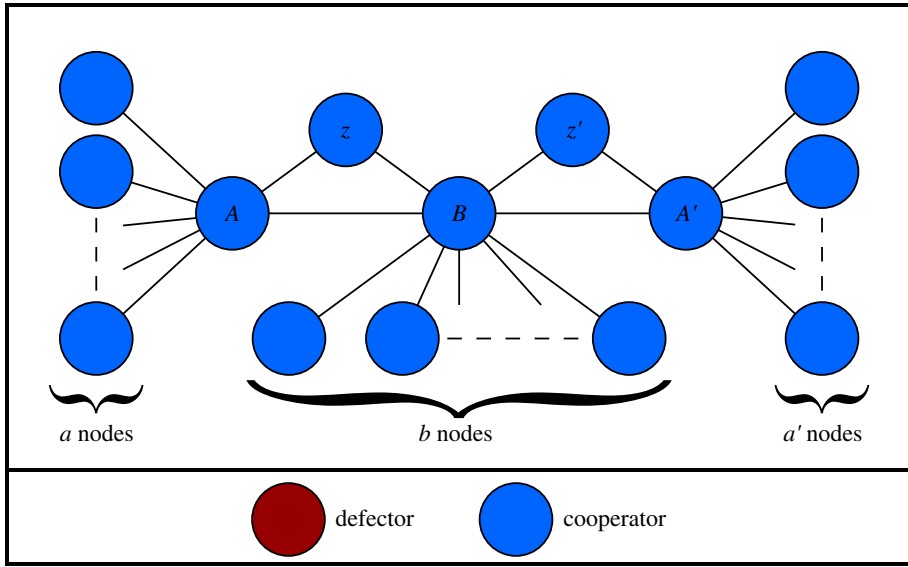

**Figure 6.** Initial graph with full cooperation.

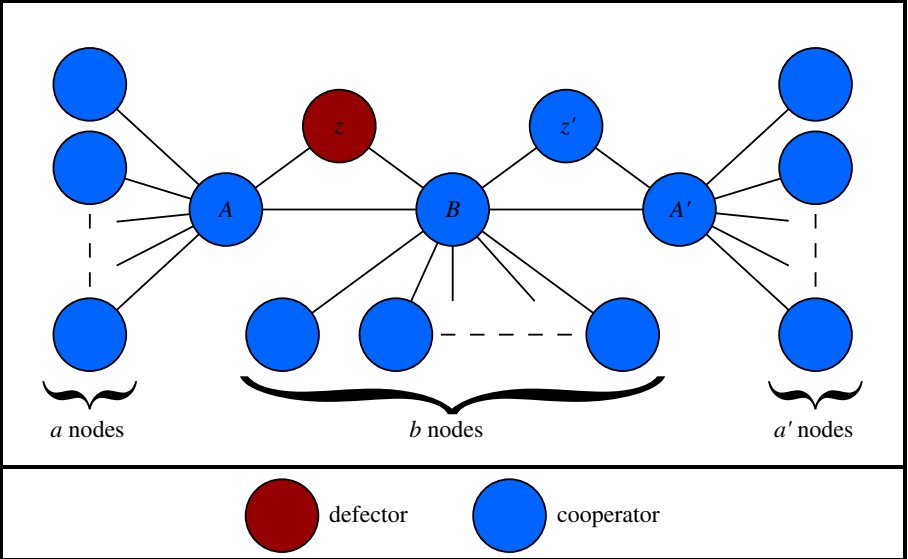

**Figure 7.** Case 1: Following time-step $t_0$.

At time-step $t_1$, the payoff of $A$ or $A'$ is bigger than that of the defector $i$ so the defector will copy their strategy and return to a cooperative state.

At the moment, we have the following payoffs:

$$P_A = R(a-1) + R \text{ (due to } B) + R \text{ (due to } z) + S \text{ (due to } i)$$
$$= R(a+1) + S$$
$$P_{A'} = R(a'-1) + R \text{ (due to } B) + R \text{ (due to } z') + S \text{ (due to } i)$$
$$= R(a'+1) + S$$
$$P_i = T \text{ (due to } A \text{ or } A') = T.$$

This results in the following constraints on the graph:

$$P_i < P_A \text{ and } P_i < P_{A'} \Leftrightarrow$$
$$T < R(a+1) + S; \text{ Already covered by (A 1)}$$
$$T < R(a'+1) + S; \text{ Already covered by (A 2).}$$

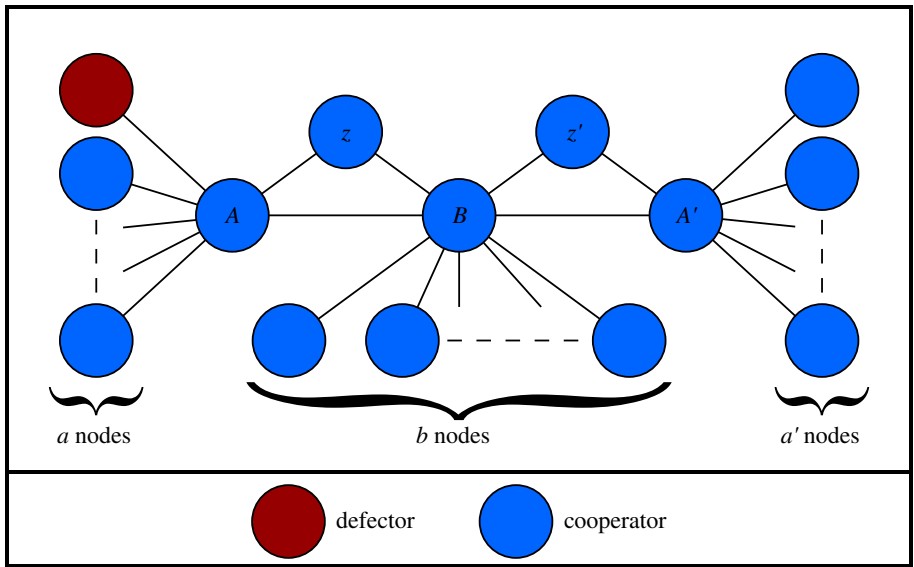

**Figure 8.** Case 2: Following time-step $t_0$ showing one of the $a$ nodes perturbed.

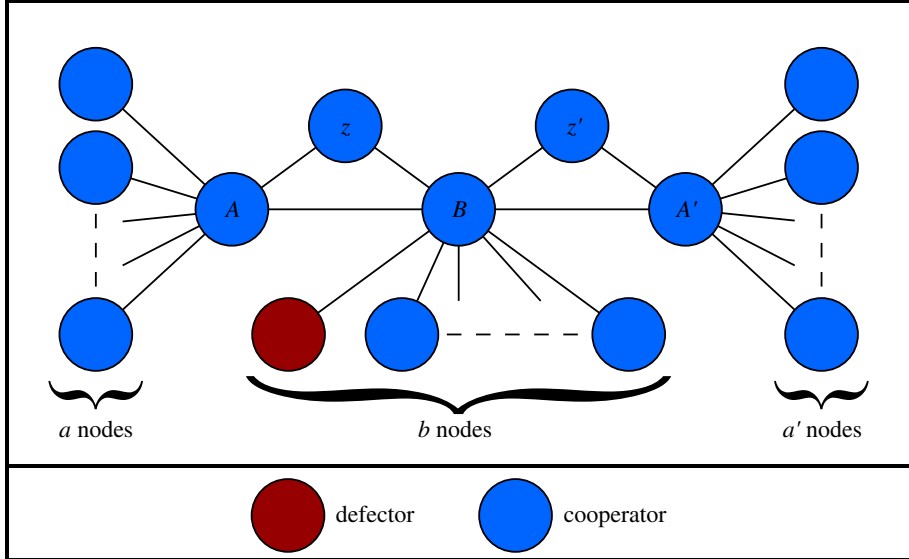

**Figure 9.** Case 3: Following time-step $t_0$ showing one of the $b$ nodes perturbed.

**Case 3: One of the $b$ nodes is perturbed**

At time-step $t_0$, the initial node, $i$, defects as shown in figure 9.

At time-step $t_1$, the payoff of $B$ is bigger than that of the initial defector so the initial defector, $i$, will return to cooperation by copying the strategy of node $B$.

At that moment, the following payoffs are rewarded:

$$P_B = R(b-1) + R \text{ (due to } A) + R \text{ (due to } z)$$
$$+ R \text{ (due to } A') + R \text{ (due to } z') + S \text{ (due to } i) = R(b+3) + S$$

and

$$P_i = T \text{ (due to } B) = T.$$

We now have the following constraints:

$P_i < P_B \Leftrightarrow$

$T < R(b+3) + S$; previously covered by (A 3), so no addition to the set of constraints.

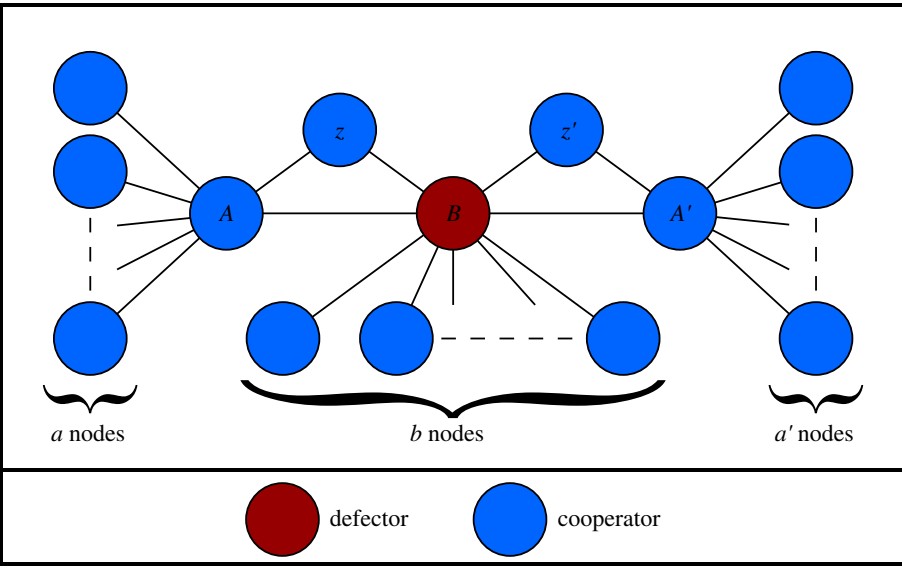

**Figure 10.** Case 4: Following time-step $t_0$ showing the $B$ node perturbed.

**Case 4: Node B is perturbed**

At time-step $t_0$, node $B$ defects as depicted in figure 10. Let $b_i$ denote any one of the $b$ nodes.

At time-step $t_1$, the payoff for $A$ and the payoff for $A'$ are both bigger than the payoff for node $B$, so node $B$ will change to cooperation. The payoff of the node $B$ is bigger than the $b$ nodes, so all the $b$ nodes will turn to defection (figure 11).

At that time-step, the following payoffs are rewarded:

$$P_B = T \times b + T \text{ (due to } A) + T \text{ (due to } A') + T \text{ (due to } z) + T \text{ (due to } z') = T(b + 4)$$
$$P_A = R \times a + R \text{ (due to } z) + S \text{ (due to } B) = R(a + 1) + S.$$

Similarly,

$$P_{A'} = R(a' + 1) + S$$
$$P_{b_i} = S \text{ (due to } B).$$

This gives the following requirements:

$$P_B < P_A \text{ and } P_B < P_{A'} \Leftrightarrow$$
$$\mathbf{T(b + 4) < R(a + 1) + S} \tag{A4}$$
$$\mathbf{T(b + 4) < R(a' + 1) + S} \tag{A5}$$

$$P_{b_i} < P_B \Leftrightarrow$$
$$S < T(b + 4); \text{ this is true as } S < T \text{ and } b > 0.$$

At time-step $t_2$, the payoff for $B$ is bigger then the payoff obtained by all the $b$ nodes, so the $b$ nodes will now return to cooperation.

At the moment, we have the following payoffs:

$$P_B = S \times b + R \text{ (due to } A) + R \text{ (due to } A') + R \text{ (due to } z) + R \text{ (due to } z') = S \times b + 4R$$
$$P_{b_i} = T \text{ (due to } B).$$

We now have the following constraints:

$$P_B > P_{b_i} \Leftrightarrow$$
$$\mathbf{S \times b + 4R > T.} \tag{A6}$$

Note that the number 4 (in $4R$) can be increased by adding more enabler nodes.

**Case 5: Node A or A′ is perturbed**

At time-step $t_0$, node $A$ is perturbed and becomes a defector (as depicted in figure 12).

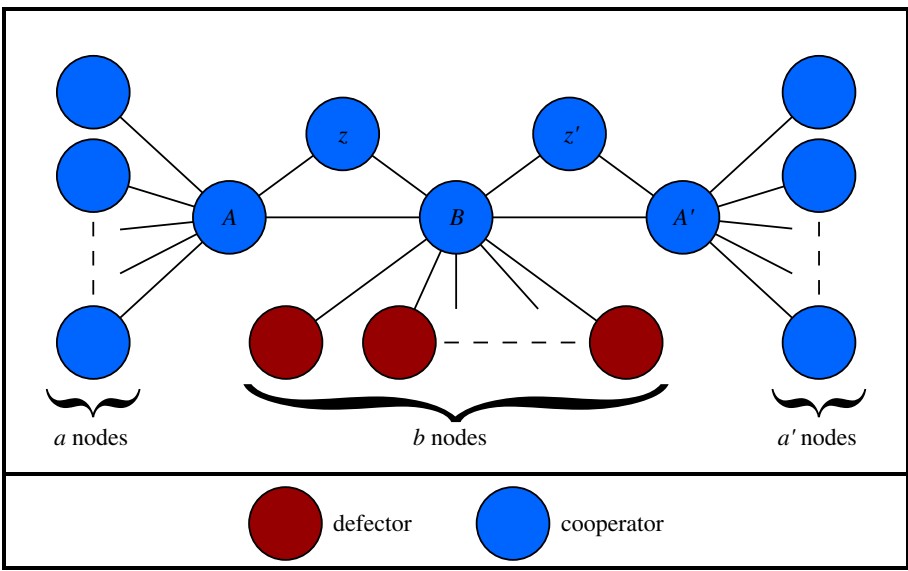

**Figure 11.** Case 4: Following time-step $t_1$.

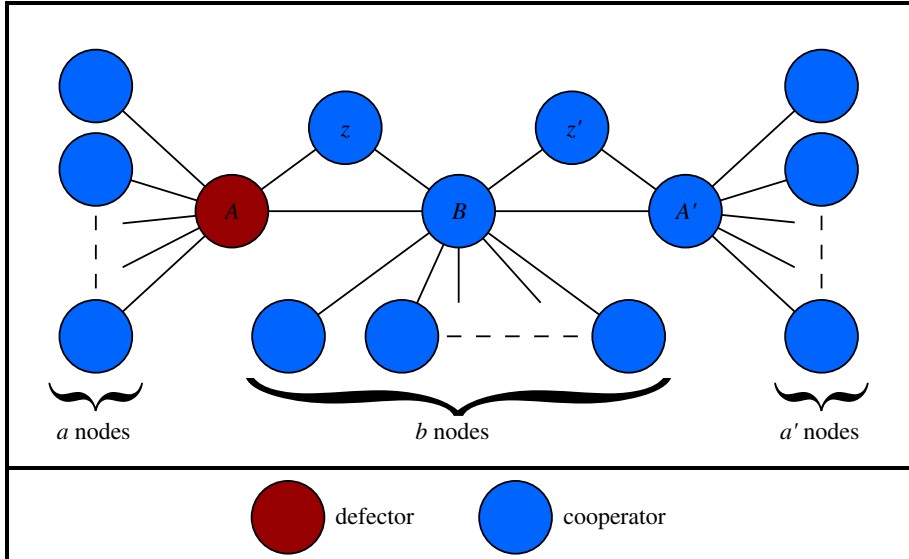

**Figure 12.** Case 5: Following time-step $t_0$ showing the $A$ node perturbed.

At time-step $t_1$, the payoff of the initial defector, $A$ (or $A'$), is bigger than all the nodes connected to it, so all the $a$ (or $a'$) nodes and the $z$ (or the $z'$) node will defect and if the payoff of the initial defector is bigger then the other critical node, the $B$ node will defect (figure 13).

At the moment, we have the following payoffs:

$$P_A \text{ (or } P_{A'}) = T \times a \text{ (or } T \times a') + T \text{ (due to } z(\text{or } z')) + T \text{ (due to } B)$$
$$= T \times (a + 2) \text{ (or } T \times (a' + 2))$$
$$P_{a_i} \text{ (or } P_{a'_i}) = S \text{ (due to } A \text{ (or } A'))$$

$$P_z \text{ (or } P_{z'}) = R \text{ (due to } B) + S \text{ (due to } A \text{ (or } A')) = R + S.$$

$$P_B = R \times b + R \text{ (due to } A' \text{ (or } A)) + R \text{ (due to } z')$$
$$+ R \text{ (due to } z) + S \text{ (due to } A \text{ (or } A')) = R(b + 3) + S$$
$$P_{A'} \text{ (or } P_A) = R \times a' \text{ (or } R \times a)$$
$$+ R \text{ (due to } z' \text{ (or due to } z)) + R \text{ (due to } B) = R(a' + 2) \text{ (or } R(a + 2)).$$

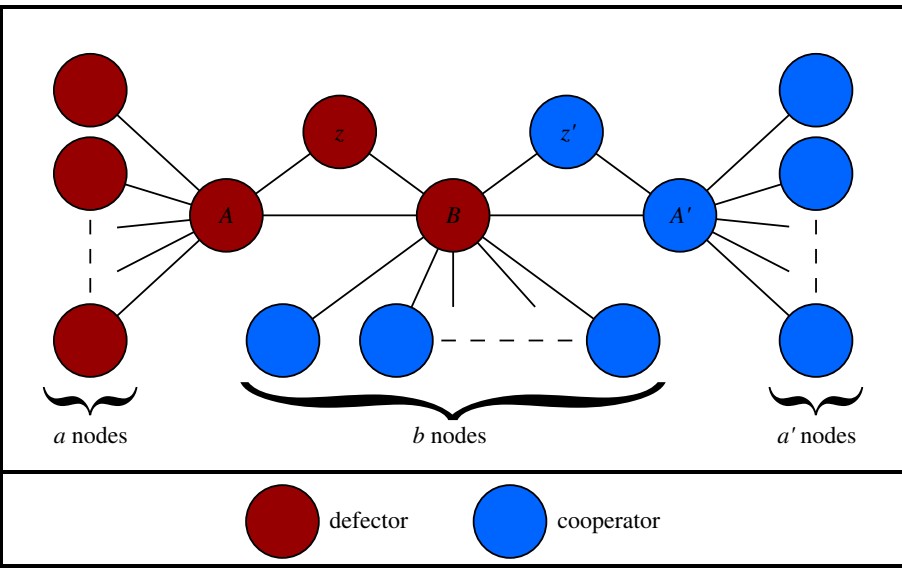

**Figure 13.** Case 5: Following time-step $t_1$.

We now have the following requirements:

$$P_A \text{ (or } P_{A'}) > P_{a_i} \text{ (or } P_{a'_i})$$

$$P_A \text{ (or } P_{A'}) > P_z \text{ (or } P_{z'})$$

$$P_A \text{ (or } P_{A'}) > P_B \Leftrightarrow$$

$\quad T(a+2) \text{ (or } T(a'+2)) > S, \text{ true as } T > S \text{ and } a \text{ and } a' > 0$

$\quad T(a+2) \text{ (or } T(a'+2)) > R+S, \text{ true as } T > S, T > R \text{ and both } a \text{ and } a' > 0$

$\quad T(a+2) \text{ (or } T(a'+2)) > R(b+3) + S, \text{ true given (A 4)},$

$\qquad T(a+2) > R(a+1) + S \text{ and } T(b+4) > R(b+3) + S.$

If $P_A$ (or $P_{A'}$) $> P_{A'}$ (or $P_A$), we will proceed to time-step $t_2$, otherwise we will jump straight to time-step $t_4$.

At time-step $t_2$, the payoff of node $A'$ is bigger then the payoff of node $B$, and the payoff of node $A'$ is bigger then the payoff of $A$. Hence, all the $b$ nodes defect, and the $B$ node cooperates (figure 14).

At the moment, we have the following payoffs:

$$P_B = T \times b + P \text{ (due to } A) + T \text{ (due to } A') + P \text{ (due to } z) + T \text{ (due to } z')$$

$$= T(b+2) + 2P$$

$$P_{A'} = R \times a' + R \text{ (due to } z') + S \text{ (due to } B) = R(a'+1) + S$$

$$P_A = P \times a + P \text{ (due to } z) + P \text{ (due to } B) = P(a+2).$$

This results in the following requirements:

$$P_B < P_{A'} \Leftrightarrow$$

$$T(b+2) + 2P < R(a'+1) + S.$$

Similarly for $A'$ as the initial defector

$$T(b+2) + 2P < R(a+1) + S.$$

These are already satisfied given (A 4) and (A 5) and given $T > P$.

$$P_{A'} > P_A \Leftrightarrow$$

$$\mathbf{R}(\mathbf{a'} + \mathbf{1}) + \mathbf{S} > \mathbf{P}(\mathbf{a} + \mathbf{2}). \tag{A 7}$$

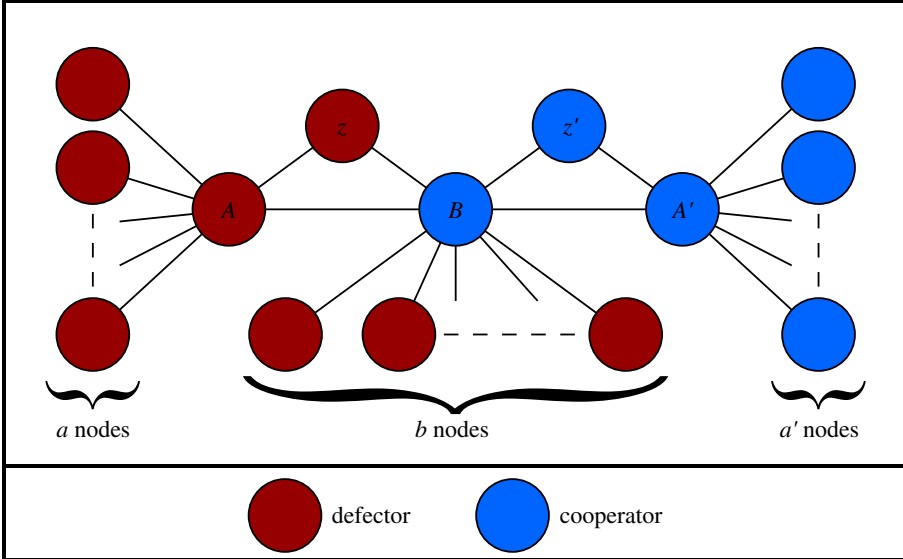

**Figure 14.** Case 5: Following time-step $t_2$.

Similarly for A′ as the initial defector

$$\mathbf{R(a+1)+S > P(a'+2)}. \tag{A 8}$$

At time-step $t_3$, the payoff of node $B$ is bigger then the payoff of all the $b$ nodes, the payoff of $A$ needs to be smaller than the payoff of A′. All the $b$ nodes will cooperate (figure 15).[2,3]

At the moment, we have the following payoffs:

$$P_B = S \times b + R \text{ (due to } A') + R \text{ (due to } z') + S \text{ (due to } A) + S \text{ (due to } z)$$
$$= S(b+2) + 2R$$
$$P_{b_i} = T \text{ (due to } B)$$
$$P_{A'} = R \times a' + R \text{ (due to } z') + R \text{ (due to } B) = R(a'+2)$$
$$P_A = P \times a + P \text{ (due to } z) + T \text{ (due to } B) = P(a+1) + T.$$

We now have the following requirements:

$$P_B > P_{b_i} \Leftrightarrow$$
$$\mathbf{S(b+2) + 2R > T} \tag{A 9}$$

$$P_A < P_{A'} \Leftrightarrow$$
$$\mathbf{P(a+1) + T < R(a'+2)}. \tag{A 10}$$

Similarly for $A'$ as the initial defector

$$P_{A'} < P_A \Leftrightarrow$$
$$\mathbf{P(a'+1) + T < R(a+2)}. \tag{A 11}$$

At time-step $t_4$, the payoffs of node $A$ and $z$ are smaller then the payoff of node $B$.
The $A$ node and the $z$ node will then cooperate (figure 16).
At that moment, we have the following payoffs:

$$P_B = R \times b + R \text{ (due to } A') + R \text{ (due to } z') + S \text{ (due to } A) + S \text{ (due to } z) = R(b+2) + 2S$$

---

[2]Some of the $b$ nodes may be allowed to have a higher degree; in this case the time-step $t_3$ will be repeated multiple times, the $b$ nodes need to be set such that all of them will cooperate eventually (the payoff of node $B$ keeps on increasing each time until all the $b$ nodes cooperate).

[3]In most cases $P_A > P_B$; when this happens the process progresses to time-step $t_4$ at the end of time-step $t_3$; otherwise if $P_B < P_A$ (when $P \approx 0$); the process proceeds to time-step $t_5$ following time-step $t_3$.

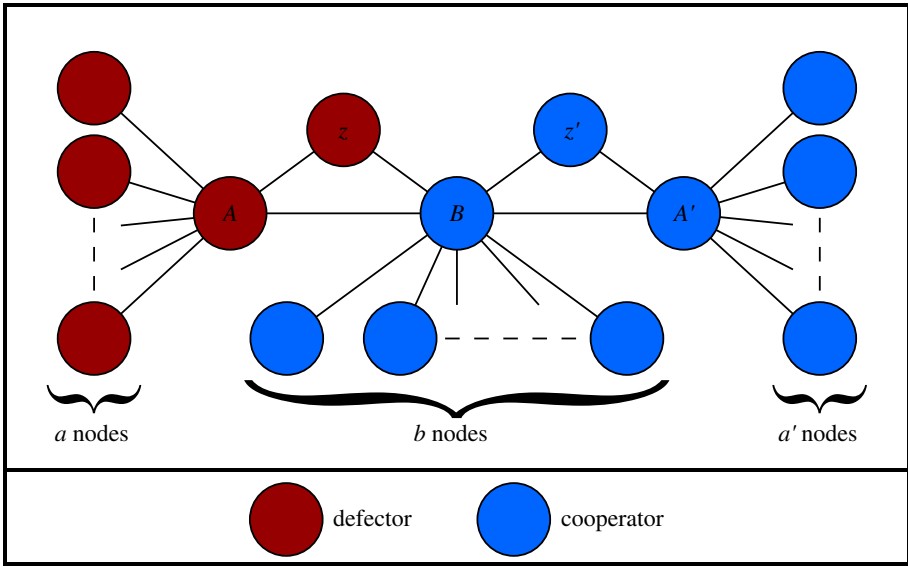

**Figure 15.** Case 5: Following time-step $t_3$.

and

$$P_A = P \times a + P \text{ (due to } z) + T \text{ (due to } B) = P(a + 1) + T.$$

We now have the following requirements:

$$P_A < P_B \Leftrightarrow$$
$$\mathbf{P(a + 1) + T < R(b + 2) + 2S}. \tag{A 12}$$

Similarly for A′ as the initial defector

$$P_{A'} < P_B \Leftrightarrow$$
$$\mathbf{P(a' + 1) + T < R(b + 2) + 2S}. \tag{A 13}$$

At time-step $t_5$, the payoff of node $A$ is bigger then the payoff of all the $a$ nodes; all the $a$ nodes will cooperate.

At the moment, we have the following payoffs:

$$P_A = S \times a + R \text{ (due to } z) + R \text{ (due to } B) = S \times a + 2R$$

and

$$P_{a_i} = T \text{ (due to } A).$$

This results in the following requirements:

$$P_{a_i} < P_A \Leftrightarrow$$
$$\mathbf{T < S \times a + 2R}. \tag{A 14}$$

Similarly for A′ as the initial defector

$$P_{a'_i} < P_{A'} \Leftrightarrow$$
$$\mathbf{T < S \times a' + 2R}. \tag{A 15}$$

Having considered all the cases and any constraints on the graph, we have the resulting list of constraints/requirements.

(A 1) $\mathbf{2T < R(a + 1) + S}$
(A 2) $\mathbf{2T < R(a'+1) + S}$
(A 3) $\mathbf{2T < R(b + 3) + S}$
(A 4) $\mathbf{T(b + 4) < R(a + 1) + S}$
(A 5) $\mathbf{T(b + 4) < R(a' + 1) + S}$

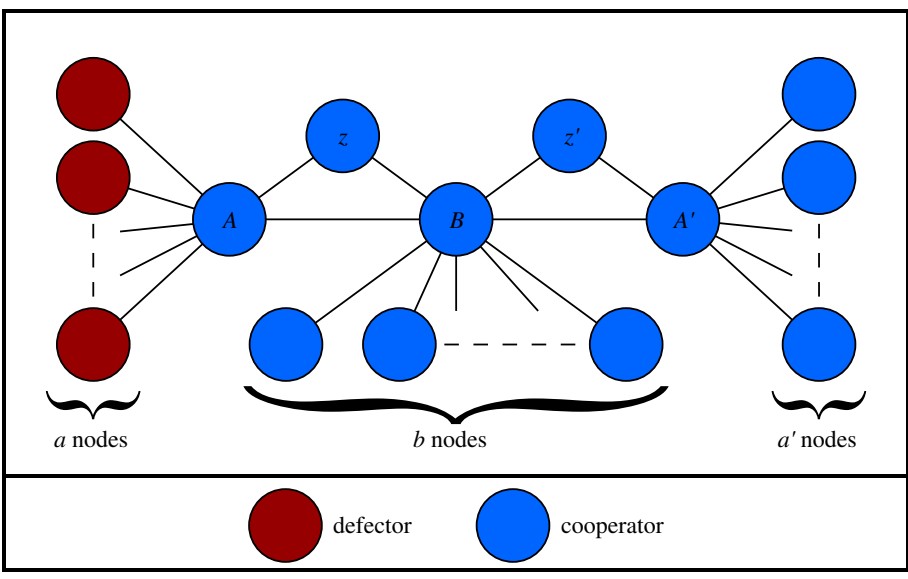

**Figure 16.** Case 5: Following at time-step $t_4$.

(A 6) $\mathbf{S \times b + 4R > T}$
(A 7) $\mathbf{R(a' + 1) + S > P(a + 2)}$
(A 8) $\mathbf{R(a + 1) + S > P(a' + 2)}$
(A 9) $\mathbf{S(b + 2) + 2R > T}$
(A 10) $\mathbf{P(a + 1) + T < R(a' + 2)}$
(A 11) $\mathbf{P(a' + 1) + T < R(a + 2)}$
(A 12) $\mathbf{P(a + 1) + T < R(b + 2) + 2S}$
(A 13) $\mathbf{P(a' + 1) + T < R(b + 2) + 2S}$
(A 14) $\mathbf{T < S \times a + 2R}$
(A 15) $\mathbf{T < S \times a' + 2R.}$

# Appendix B

## B.1. Extra requirement for the graph

For the definition of the critical nodes, in the case where it and all its minor nodes cooperate, it cannot be influenced to defect, to be fully correct, we need to consider one more scenario; the scenario where both the $B$ node and the enabler node connecting them defect.

**Case 6: Node $B$ is perturbed and node $z$ or $z'$ is perturbed**

At time-step $t_0$, node $B$ and $z$ or $z'$ defect (figure 17).

At time-step $t_1$, the payoffs of $A$ and $A'$ are bigger then the payoff of both node $B$ and $z$ or $z'$, so node $B$ and $z$ or $z'$ will cooperate. The payoff of node $B$ is bigger than the $b$ nodes, so all the $b$ nodes will defect.

At the moment, we have the following payoffs:

$$P_B = T \times b + T \text{ (due to } A) + T \text{ (due to } A') + P \text{ (due to } z) + T \text{ (due to } z')$$
$$= T(b + 3) + P$$
$$P_A = R \times a + S \text{ (due to } z) + S \text{ (due to } B) = R \times a + 2S$$

Similarly for $z'$ as the initial defector, $P_{A'} = R(a'+1) + S$

$$P_{b_i} = S \text{ (due to } B)$$

$$P_z = T \text{ (due to } A) + P \text{ (due to } B)$$

We now have the following requirements:

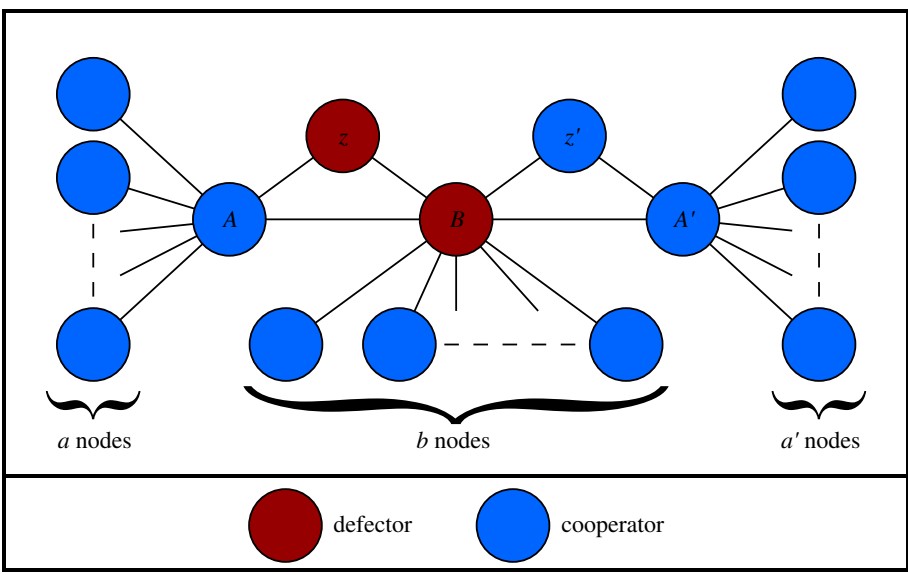

**Figure 17.** Case 6: Following time-step $t_0$ showing the $B$ node and $z$ node perturbed.

$$P_z < P_A \Leftrightarrow$$

$$\mathbf{T + P < R \times a + 2S}. \tag{B1}$$

Similarly for $z'$ as the initial defector

$$P_{z'}, \ < P_{A'} \Leftrightarrow$$

$$\mathbf{T + P < R \times a' + 2S}. \tag{B2}$$

$$P_{b_i} < P_B \Leftrightarrow$$
$$S < T(b + 3) + P, \ \text{true since } S < T \text{ and } b > 0$$
$$P_B < P_A \Leftrightarrow$$

$$\mathbf{T(b + 3) + P < R \times a + 2S}. \tag{B3}$$

Similarly for $z'$ as the initial defector

$$P_B < P_{A'} \Leftrightarrow$$

$$\mathbf{T(b + 3) + P < R \times a' + 2S}. \tag{B4}$$

At the next time-step, we will be in the same spot as case 4, time-step $t_1$.

# Appendix C

## C.1. Examining the requirements of the Locodi graph

**List of requirements:**

(A 1) **$2T < R(a + 1) + S$**
(A 2) **$2T < R(a'+1)+S$**
(A 3) **$2T < R(b + 3) + S$**
(A 4) **$T(b + 4) < R(a + 1) + S$**
(A 5) **$T(b + 4) < R(a' + 1)+S$**
(A 6) **$S \times b + 4R > T$**
(A 7) **$R(a'+1) + S > P(a + 2)$**
(A 8) **$R(a + 1) + S > P(a' + 2)$**
(A 9) **$S(b + 2) + 2R > T$**
(A 10) **$P(a + 1) + T < R(a' + 2)$**

(A 11) $P(a' + 1) + T < R(a + 2)$

(A 12) $P(a + 1) + T < R(b + 2) + 2S$

(A 13) $P(a' + 1) + T < R(b + 2) + 2S$

(A 14) $T < S \times a + 2R$

(A 15) $T < S \times a' + 2R$

(B 1) $T + P < R \times a + 2S$

(B 2) $T + P < R \times a' + 2S$

(B 3) $T(b + 3) + P < R \times a + 2S$

(B 4) $T(b + 3) + P < R \times a' + 2S.$

From (A 9):

$S(b + 2) + 2R > T \Leftrightarrow \mathbf{b} > (\mathbf{T - 2R - 2S})/\mathbf{S}$, if $S > 0$; if $S = 0 \Leftrightarrow \mathbf{2R > T}$ (we can increase the number 2 in $2R$ by increasing the number of enabler nodes)

From (A 6):

$S \times b + 4R > T \Leftrightarrow b > (T - 4R)/S$, if $S > 0$; if $S = 0 \Leftrightarrow 4R > T$

Note: this is irrelevant due to (A 9) (because both have to be true and (A 6) is always true when (A 9) is true).

From (A 14):

$T < S \times a + 2R \Leftrightarrow \mathbf{a} > (\mathbf{T - 2R})/\mathbf{S}$, if $S > 0$; if $S = 0 \Leftrightarrow \mathbf{2R > T}$

From (A 15):

$T < S \times a' + 2R \Leftrightarrow \mathbf{a'} > (\mathbf{T - 2R})/\mathbf{S}$, if $S > 0$; if $S = 0 \Leftrightarrow \mathbf{2R > T}$

From (A 1), (A 2), (A 7), (A 8), (A 10), (A 11), (B 1) and (B 2):

(A 1): $2T < R(a + 1) + S \Leftrightarrow$

$$a > \frac{2T - S - R}{R}$$

(A 2): $2T < R(a' + 1) + S \Leftrightarrow$

$$a' > \frac{2T - S - R}{R}$$

(A 7): $R(a' + 1) + S > P(a + 2) \Leftrightarrow$

$$a < \frac{R(a' + 1) + S - 2P}{P}$$
$$a' > \frac{P(a + 2) - S - R}{R}$$

(A 8): $R(a + 1) + S > P(a' + 2) \Leftrightarrow$

$$a' < \frac{R(a + 1) + S - 2P}{P}$$
$$a > \frac{P(a' + 2) - S - R}{R}$$

(A 10): $P(a + 1) + T < R(a' + 2) \Leftrightarrow$

$$a < \frac{R(a' + 2) - T - P}{P}$$
$$a' > \frac{P(a + 1) + T - 2R}{R}$$

(A 11): $P(a' + 1) + T < R(a + 2) \Leftrightarrow$

$$a' < \frac{R(a + 2) - T - P}{P}$$
$$a > \frac{P(a' + 1) + T - 2R}{R}$$

(B 1): $T + P < R \times a + 2S \Leftrightarrow$

$$a > \frac{T + P - 2S}{R}$$

(B 2): $T + P < R \times a' + 2S \Leftrightarrow$

$$a' > \frac{T + P - 2S}{R}$$

(A 1), (A 2), (A 7), (A 8), (A 10), (A 11), (B 1) and (B 2): $\Leftrightarrow$

$$\max\left(\frac{2T - S - R}{R}, \frac{P(a' + 2) - S - R}{R}, \frac{P(a' + 1) + T - 2R}{R}, \frac{T + P - 2S}{R}\right) < a$$
$$< \min\left(\frac{R(a' + 1) + S - 2P}{P}, \frac{R(a' + 2) - T - P}{P}\right)$$
$$\max\left(\frac{2T - S - R}{R}, \frac{P(a + 2) - S - R}{R}, \frac{P(a + 1) + T - 2R}{R}, \frac{T + P - 2S}{R}\right) < a'$$
$$< \min\left(\frac{R(a + 1) + S - 2P}{P}, \frac{R(a + 2) - T - P}{P}\right)$$

From (A 3), (A 4), (A 5), (A 12), (A 13), (B 3) and (B 4):
(A 3): $2T < R(b + 3) + S \Leftrightarrow$

$$b > \frac{2T - S - 3R}{R}$$

(A 4): $T(b + 4) < R(a + 1) + S \Leftrightarrow$

$$b < \frac{R(a + 1) + S - 4T}{T}$$
$$a > \frac{T(b + 4) - S - R}{R}$$

(A 5): $T(b + 4) < R(a' + 1) + S \Leftrightarrow$

$$b < \frac{R(a' + 1) + S - 4T}{T}$$
$$a' > \frac{T(b + 4) - S - R}{R}$$

(A 12): $P(a + 1) + T < R(b + 2) + 2S \Leftrightarrow$

$$b > \frac{P(a + 1) + T - 2S - 2R}{R}$$
$$a < \frac{R(b + 2) + 2S - T - P}{P}$$

(A 13): $P(a' + 1) + T < R(b + 2) + 2S \Leftrightarrow$

$$b > \frac{P(a' + 1) + T - 2S - 2R}{R}$$
$$a' < \frac{R(b + 2) + 2S - T - P}{P}$$

(B 3): $T(b + 3) + P < R \times a + 2S \Leftrightarrow$

$$b < \frac{R \times a + 2S - P - 3T}{T}$$
$$a > \frac{T(b + 3) + P - 2S}{R}$$

(B 4): $T(b + 3) + P < R \times a' + 2S \Leftrightarrow$

$$b < \frac{R \times a' + 2S - P - 3T}{T}$$
$$a' > \frac{T(b + 3) + P - 2S}{R}$$

(A 3), (A 4), (A 5), (A 12), (A 13), (B 3) and (B 4): ⇔

$$\max\left(\frac{2T - S - 3R}{R}, \frac{P(a+1) + T - 2S - 2R}{R}, \frac{P(a'+1) + T - 2S - 2R}{R}\right) < b$$

$$< \min\left(\frac{R(a+1) + S - 4T}{T}, \frac{R(a'+1) + S - 4T}{T},\right.$$

$$\left.\frac{R \times a + 2S - P - 3T}{T}, \frac{R \times a' + 2S - P - 3T}{T}\right)$$

As the values $a$, $b$, $a'$ tend to infinity, we get the following requirement $P/R < R/T$

$$\max\left(\frac{T(b+4) - S - R}{R}, \frac{T(b+3) + P - 2S}{R}\right) < a, a' < \frac{R(b+2) + 2S - T - P}{P}$$

Hence, we get the following requirement $T/R < R/P$ which is the same as $P/R < R/T$.

From $P/R < R/T \Leftrightarrow P < R^2/T$ and $T < R^2/P$, we have the following relation between payoffs: $\mathbf{R^2/P} > T > R > \mathbf{R^2/T} > P > S$.

# Appendix D

## D.1. Smallest possible graph (classical Prisoner's Dilemma payoffs) that guarantees robustness to one perturbation

We have enumerated the different cases for each of the different types of perturbation and illustrated how recovery to cooperation occurs. In doing so, we specified the various constraints that need to apply in the graph. These constraints allow us to derive the minimal size of the graph that will guarantee robustness.

We have the following payoffs: $T = 5$, $R = 3$, $P = 1$ and $S = 0$.
From appendix C, we have: max $((T(b+4) - S - R)/R, (T(b+3) + P - 2S)/R) < a, a' < (R \times (b+2) + 2S - T - P)/P \Leftrightarrow \max((5 \times (b+4) - 0 - 3)/3, (5 \times (b+3) + 1 - 0)/3) < a, a' < (3 \times (b+2) + 2 \times 0 - 5 - 1)/1 \Leftrightarrow (5b + 17)/3 < a, a' < 3 \times b$
We need to find the smallest value of $b$ where this is possible: $(5b + 17)/3 < 3b \Leftrightarrow 5b + 17 < 9b \Leftrightarrow 17 < 4b \Leftrightarrow b > 4.25$, $b$ is a natural number $\Leftrightarrow b >= 5$
For $b = 5$, we have: $(5 \times 5 + 17)/3 < a, a' < 3 \times 5 \Leftrightarrow 14 < a, a' < 15$, $a, a'$ are natural numbers $\Leftrightarrow b > 5$
For $b = 6$, we have: $(5 \times 6 + 17)/3 < a, a' < 3 \times 6 \Leftrightarrow 15.(6) < a, a' < 18 \Leftrightarrow$ smallest value for $a, a'$ is 16

**Final check to verify that $a = 16$, $a' = 16$ and $b = 6$ respect all the requirements.**

(A 1) $\mathbf{2T < R(a + 1) + S} \Leftrightarrow 2 \times 5 < 3 \times (16 + 1) + 0 \Leftrightarrow 10 < 51$ (TRUE)
(A 2) $\mathbf{2T < R(a' + 1) + S} \Leftrightarrow 2 \times 5 < 3 \times (16 + 1) + 0 \Leftrightarrow 10 < 51$ (TRUE)
(A 3) $\mathbf{2T < R(b + 3) + S} \Leftrightarrow 2 \times 5 < 3 \times (6 + 1) + 0 \Leftrightarrow 10 < 21$ (TRUE)
(A 4) $\mathbf{T(b + 4) < R(a + 1) + S} \Leftrightarrow 5 \times (6 + 4) < 3 \times (16 + 1) + 0 \Leftrightarrow 50 < 51$ (TRUE)
(A 5) $\mathbf{T(b + 4) < R(a' + 1) + S} \Leftrightarrow 5 \times (6 + 4) < 3 \times (16 + 1) + 0 \Leftrightarrow 50 < 51$ (TRUE)
(A 6) $\mathbf{S \times b + 4R > T} \Leftrightarrow 0 \times 6 + 4 \times 3 > 5 \Leftrightarrow 12 > 5$ (TRUE)
(A 7) $\mathbf{R(a' + 1) + S > P(a + 2)} \Leftrightarrow 3 \times (16 + 1) + 0 > 16 + 2 \Leftrightarrow 51 > 18$ (TRUE)
(A 8) $\mathbf{R(a + 1) + S > P(a' + 2)} \Leftrightarrow 3 \times (16 + 1) + 0 > 16 + 2 \Leftrightarrow 51 > 18$ (TRUE)
(A 9) $\mathbf{S(b + 2) + 2R > T} \Leftrightarrow 0 \times (6 + 2) + 2 \times 3 > 5 \Leftrightarrow 6 > 5$ (TRUE)
(A 10) $\mathbf{P(a + 1) + T < R(a' + 2)} \Leftrightarrow 1 \times (16 + 1) + 5 < 3 \times (16 + 2) \Leftrightarrow 22 < 54$ (TRUE)
(A 11) $\mathbf{P(a' + 1) + T < R(a + 2)} \Leftrightarrow 1 \times (16 + 1) + 5 < 3 \times (16 + 2) \Leftrightarrow 22 < 54$ (TRUE)
(A 12) $\mathbf{P(a + 1) + T < R(b + 2) + 2S} \Leftrightarrow 1 \times (16 + 1) + 5 < 3 \times (6 + 2) + 2 \times 0 \Leftrightarrow 22 < 24$ (TRUE)
(A 13) $\mathbf{P(a' + 1) + T < R(b + 2) + 2S} \Leftrightarrow 1 \times (16 + 1) + 5 < 3 \times (6 + 2) + 2 \times 0 \Leftrightarrow 22 < 24$ (TRUE)
(A 14) $\mathbf{T < S \times a + 2R} \Leftrightarrow 5 < 0 \times 16 + 2 \times 3 \Leftrightarrow 5 < 6$ (TRUE)
(A 15) $\mathbf{T < S \times a' + 2R} \Leftrightarrow 5 < 0 \times 16 + 2 \times 3 \Leftrightarrow 5 < 6$ (TRUE)
(B 1) $\mathbf{T + P < R \times a + 2S} \Leftrightarrow 5 + 1 < 3 \times 16 + 2 \times 0 \Leftrightarrow 6 < 48$ (TRUE)
(B 2) $\mathbf{T + P < R \times a' + 2S} \Leftrightarrow 5 + 1 < 3 \times 16 + 2 \times 0 \Leftrightarrow 6 < 48$ (TRUE)
(B 3) $\mathbf{T(b + 3) + P < R \times a + 2S} \Leftrightarrow 5 \times (6 + 3) + 1 < 3 \times 16 + 2 \times 0 \Leftrightarrow 46 < 48$ (TRUE)
(B 4) $\mathbf{T(b + 3) + P < R \times a' + 2S} \Leftrightarrow 5 \times (6 + 3) + 1 < 3 \times 16 + 2 \times 0 \Leftrightarrow 46 < 48$ (TRUE)

Since all the requirements are true the smallest graph has $a = 16$, $a' = 16$ and $b = 6$.

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
