## [Peer Review File · Royal Society Open Science]

Review History

RSOS-201958.R0 (Original submission)

Review form: Reviewer 1

Is the manuscript scientifically sound in its present form?

No

Are the interpretations and conclusions justified by the results?

No

Is the language acceptable?

Yes

Do you have any ethical concerns with this paper?

No

Have you any concerns about statistical analyses in this paper?

No

Recommendation?

Major revision is needed (please make suggestions in comments)

Comments to the Author(s)

In this work the authors study the effect of network topology on the maintenance of cooperation in evolutionary (social dilemma) game with players located on a particular graph. Namely, the numerical analyses are restricted to networks exhibiting one-dimensional character. The evolution of strategy distribution is controlled by the imitation of the best neighbour. For this model the authors describe some simple (and well-known) phenomena supporting the dominance of cooperation in the final stationary state. I cannot recommend the publication of this work because of the absence of sufficiently important and new results.

Detailed critical remarks

- 1) The most relevant topological features (irregularity in the number of neighbours, overlapping small cliques/triangles or comets versus stars) - that support the spreading of cooperation - are well investigated in the literature and surveyed in several reviews. Unfortunately, the authors don't use this knowledge in the interpretation of their results.
- 2) The chain-like topological structure of connections and the applied dynamical rule are not realistic. I think that the scientific value of this work can be improved by considering the effect of stochastic imitations, too. The one-dimensional feature of this connectivity structures can be exploited by determining the average velocity of the boundary (separating homogeneous regions of cooperation and defection) for different parameters.
- 3) The descriptions of the model and method are verbose. A more concise survey of these details would improve the presentation.

Review form: Reviewer 2

Is the manuscript scientifically sound in its present form?

No

Are the interpretations and conclusions justified by the results?

Yes

Is the language acceptable?

Yes

Do you have any ethical concerns with this paper?

No

Have you any concerns about statistical analyses in this paper?

No

Recommendation?

Major revision is needed (please make suggestions in comments)

Comments to the Author(s)

In this manuscript, the authors begin with a fully cooperative population and explore the robustness of the population to the introduction of defectors. They introduce a graph structure

that has the property that the initial fully cooperative population is robust to any perturbation. They present a proof of the property and specify the necessary constraints on the graph. After reading the manuscript, I have found it interesting, but I have some following questions or comments on this work.

(1) In this work, the authors use a simple strategy update mechanism where agents copy the best performing strategy of their neighborhood. I think that the graph topology for robust cooperation the author introduced is sensitive to this strategy update mechanism. Is that true? I wonder whether the graph topology still has the property when other strategy update mechanisms are considered.

(2) In this work, the authors consider the synchronous updates under which all the players calculate their payoffs and potentially update their strategy at the same time. I wonder whether their conclusion is still valid when the asynchronous updates are considered.

Review form: Reviewer 3

Is the manuscript scientifically sound in its present form?

Yes

Are the interpretations and conclusions justified by the results?

Yes

Is the language acceptable?

Yes

Do you have any ethical concerns with this paper?

No

Have you any concerns about statistical analyses in this paper?

No

Recommendation?

Major revision is needed (please make suggestions in comments)

Comments to the Author(s)

(Note: Apologies for the brevity of this review, which is due to lockdown time constraints and the resulting home-schooling. Please be reassured that the paper was fully read even if my comments are terse.)

The paper considers an interesting topic, namely the robustness of a population situated on an underlying network topology to defection, in the context of the prisoner's dilemma. The paper presents a graph structure which, assuming that the population is initially fully cooperative, is robust to changing a single cooperator to a defector. The authors then show how this graph can be extended through the addition of nodes which adhere to a particular structure.

The problem is clearly stated, and the paper is generally well written. However, there are number of assumptions and simplifications which would benefit from clearer justification or discussion. As a result of the simplifying assumptions there are a number of open questions, some of which impact on generality of the conclusions, and the proposed graph structure.

Specific questions are as follows ("*" denotes a more significant issue):

- Is it reasonable to assuming perfect information/observation in terms of observing the best neighbouring payoffs and subsequently copying strategies? What is the impact of noise in these aspects?

* Since each node plays a game with each of its neighbours the number of edges determines the available payoff. Thus, the "hub" nodes will clearly be more influential, not just because more neighbours can potentially copy them, but because in a broadly cooperative environment it is the hubs that received the high (cumulative) payoffs in a given round. What would be the impact of copying based on "average interaction payoff", which would better reflect the relative effectiveness of the strategies?

- The comparison against the star topology is informative, but it would also be useful to consider some more realistic topologies.

- The method of extended the graph by connecting in a "line topology" seems overly restrictive. Have other extensions been considered? Given the discussion on the characteristics of a star topology it seems that this form of extension is also possible.

* When considering the introduction of defectors, what is the impact of more targeted perturbations (rather than random). For example, in the context of influence maximisation (which although rather different has parallels in terms of "strategies" potentially spreading in a connected topology), it is common to use simple targeting such as degree or degree-discount. What would be the impact of such interventions here? For example, is cooperation still achieved if the A nodes or B nodes are targeted? Graphs such as that in Figure 2, would be robust to 2 "a" nodes being perturbed, but the situation of perturbing A and A' is not clearly discussed. Some further analysis of the cases that are not cooperative in Figure 5 would be informative here.

- How does the proposed graph structure relate to structures observed in more general synthetic graphs (e.g. random, small-world, scale-free etc.) and real-world graphs (e.g. those available from the Stanford Network Analysis Project)? Are subgraphs, or motifs, of the proposed form observed elsewhere? Does their introduction add robustness to existing graphs?

* The proposed structure is related to other graph structures and motifs which have similar effects, and this needs further discussion in the related work section. For example, in the norm emergence literature there is a notion of "self reinforcing" structure, which has a similar effect to the proposed graph structure (see <https://doi.org/10.1145/2451248.2451250> and <https://dl.acm.org/doi/10.5555/2283396.2283465>).

* The phrasing "we define the following category of graphs, 'resilient graphs'" is a little misleading, since there is no formal definition and the only examples discussed are the proposed structure and a complete graph. If this really is a new category of graph more discussion is needed.

* Section 5 refers to "our robustness measure", but I don't see a clear definition of what this is.

- I suggest rephrasing the future work to be more general, rather than identifying areas such as "robustness of $k = 4$ n by n lattice graphs".

- Finally, the balance between the main body of the paper and the appendices might be improved by moving some of the technical content from the appendices into the main body.

Review form: Reviewer 4

Is the manuscript scientifically sound in its present form?

Yes

Are the interpretations and conclusions justified by the results?

Yes

Is the language acceptable?

Yes

Do you have any ethical concerns with this paper?

No

Have you any concerns about statistical analyses in this paper?

No

Recommendation?

Accept with minor revision (please list in comments)

Comments to the Author(s)

This paper introduces a graph topology that promotes the evolutionary stability of cooperation under a simple imitation learning rule. To my knowledge, the topology introduced is novel, and therefore the paper is suitable for publication by the criteria of Royal Society Open Science. I have only one major comment, and several minor comments, the addressing of which I believe will benefit the paper.

MAJOR COMMENT

In their introduction, the authors mention the extensive literature on graph structures that promote the evolution of cooperation. I was therefore surprised to see no discussion of how the graph structure introduced in this paper relates to those elsewhere in the literature. What properties does this topology share in common with others that promote cooperation? Do the authors believe these common features to be necessary for a graph to promote the evolution of cooperation?

MINOR COMMENTS

- In the learning model employed by the authors, a focal node changes its strategy to that of the highest-payoff neighboring node, if the payoff of that node is greater than the current payoff to the focal node. In the literature, two relevant payoff specifications have been considered: (i) a node's average/expected payoff (averaged over all possible interactions with neighbors), and (ii) a node's total payoff (so that nodes with more neighbors have the opportunity of a higher payoff). The authors employ specification (ii), which should be explicitly stated in the model setup, since the alternative specification of average payoffs is also reasonable (and common). Perhaps the authors could also discuss the importance of this specification for their results?

- In Figure 5, the 100-x line for the star is exact, and need not be plotted for simulated data. From a state of all cooperation on the star graph, if some nodes are subsequently changed to defection, then cooperation will eventually be restored if and only if the central node was not changed to a defector. If x percent of nodes are changed to defection, then the percentage chance that the central node was not among them is 100-x.

- Especially in the Appendix, it would be useful to give the labels for the nodes in the figures, so that one doesn't have to continually refer to Figure 2 in the main text to remind oneself which are the z , z' , etc. nodes.

Decision letter (RSOS-201958.R0)

Dear Dr O'Riordan

The Editors assigned to your paper RSOS-201958 "Introducing a graph topology for robust cooperation" have now received comments from reviewers and would like you to revise the paper in accordance with the reviewer comments and any comments from the Editors. Please note this decision does not guarantee eventual acceptance.

Please submit your revised manuscript and required files (see below) no later than 21 days from today's (ie 22-Jan-2021) date. Note: the ScholarOne system will 'lock' if submission of the revision is attempted 21 or more days after the deadline. If you do not think you will be able to meet this deadline please contact the editorial office immediately.

on behalf of Dr Derek Abbott (Associate Editor) and Marta Kwiatkowska (Subject Editor)

Reviewer comments to Author:

Reviewer: 1

Comments to the Author(s)

In this work the authors study the effect of network topology on the maintenance of cooperation in evolutionary (social dilemma) game with players located on a particular graph. Namely, the numerical analyses are restricted to networks exhibiting one-dimensional character. The evolution of strategy distribution is controlled by the imitation of the best neighbour. For this model the authors describe some simple (and well-known) phenomena supporting the dominance of cooperation in the final stationary state. I cannot recommend the publication of this work because of the absence of sufficiently important and new results.

Detailed critical remarks

- 1) The most relevant topological features (irregularity in the number of neighbours, overlapping small cliques/triangles or comets versus stars) - that support the spreading of cooperation - are well investigated in the literature and surveyed in several reviews. Unfortunately, the authors don't use this knowledge in the interpretation of their results.
- 2) The chain-like topological structure of connections and the applied dynamical rule are not realistic. I think that the scientific value of this work can be improved by considering the effect of stochastic imitations, too. The one-dimensional feature of this connectivity structures can be exploited by determining the average velocity of the boundary (separating homogeneous regions of cooperation and defection) for different parameters.
- 3) The descriptions of the model and method are verbose. A more concise survey of these details would improve the presentation.

Reviewer: 2

Comments to the Author(s)

In this manuscript, the authors begin with a fully cooperative population and explore the robustness of the population to the introduction of defectors. They introduce a graph structure that has the property that the initial fully cooperative population is robust to any perturbation. They present a proof of the property and specify the necessary constraints on the graph. After reading the manuscript, I have found it interesting, but I have some following questions or comments on this work.

- (1) In this work, the authors use a simple strategy update mechanism where agents copy the best performing strategy of their neighborhood. I think that the graph topology for robust cooperation the author introduced is sensitive to this strategy update mechanism. Is that true? I wonder whether the graph topology still has the property when other strategy update mechanisms are considered.
- (2) In this work, the authors consider the synchronous updates under which all the players calculate their payoffs and potentially update their strategy at the same time. I wonder whether their conclusion is still valid when the asynchronous updates are considered.

Reviewer: 3

Comments to the Author(s)

(Note: Apologies for the brevity of this review, which is due to lockdown time constraints and the resulting home-schooling. Please be reassured that the paper was fully read even if my comments are terse.)

The paper considers an interesting topic, namely the robustness of a population situated on an underlying network topology to defection, in the context of the prisoner's dilemma. The paper presents a graph structure which, assuming that the population is initially fully cooperative, is robust to changing a single cooperator to a defector. The authors then show how this graph can be extended through the addition of nodes which adhere to a particular structure.

The problem is clearly stated, and the paper is generally well written. However, there are number of assumptions and simplifications which would benefit from clearer justification or discussion. As a result of the simplifying assumptions there are a number of open questions, some of which impact on generality of the conclusions, and the proposed graph structure.

Specific questions are as follows ("*" denotes a more significant issue):

- Is it reasonable to assuming perfect information/observation in terms of observing the best neighbouring payoffs and subsequently copying strategies? What is the impact of noise in these aspects?

* Since each node plays a game with each of its neighbours the number of edges determines the available payoff. Thus, the "hub" nodes will clearly be more influential, not just because more neighbours can potentially copy them, but because in a broadly cooperative environment it is the hubs that received the high (cumulative) payoffs in a given round. What would be the impact of copying based on "average interaction payoff", which would better reflect the relative effectiveness of the strategies?

- The comparison against the star topology is informative, but it would also be useful to consider some more realistic topologies.

- The method of extended the graph by connecting in a "line topology" seems overly restrictive. Have other extensions been considered? Given the discussion on the characteristics of a star topology it seems that this form of extension is also possible.

* When considering the introduction of defectors, what is the impact of more targeted perturbations (rather than random). For example, in the context of influence maximisation (which although rather different has parallels in terms of "strategies" potentially spreading in a connected topology), it is common to use simple targeting such as degree or degree-discount. What would be the impact of such interventions here? For example, is cooperation still achieved if the A nodes or B nodes are targeted? Graphs such as that in Figure 2, would be robust to 2 "a" nodes being perturbed, but the situation of perturbing A and A' is not clearly discussed. Some further analysis of the cases that are not cooperative in Figure 5 would be informative here.

- How does the proposed graph structure relate to structures observed in more general synthetic graphs (e.g. random, small-world, scale-free etc.) and real-world graphs (e.g. those available from the Stanford Network Analysis Project)? Are subgraphs, or motifs, of the proposed form observed elsewhere? Does their introduction add robustness to existing graphs?

* The proposed structure is related to other graph structures and motifs which have similar effects, and this needs further discussion in the related work section. For example, in the norm emergence literature there is a notion of "self reinforcing" structure, which has a similar effect to the proposed graph structure (see <https://doi.org/10.1145/2451248.2451250> and <https://dl.acm.org/doi/10.5555/2283396.2283465>).

* The phrasing "we define the following category of graphs, 'resilient graphs'" is a little misleading, since there is no formal definition and the only examples discussed are the proposed structure and a complete graph. If this really is a new category of graph more discussion is needed.

* Section 5 refers to "our robustness measure", but I don't see a clear definition of what this is.

- I suggest rephrasing the future work to be more general, rather than identifying areas such as "robustness of $k = 4n$ by n lattice graphs".

- Finally, the balance between the main body of the paper and the appendices might be improved by moving some of the technical content from the appendices into the main body.

Reviewer: 4

Comments to the Author(s)

This paper introduces a graph topology that promotes the evolutionary stability of cooperation under a simple imitation learning rule. To my knowledge, the topology introduced is novel, and therefore the paper is suitable for publication by the criteria of Royal Society Open Science. I have only one major comment, and several minor comments, the addressing of which I believe will benefit the paper.

MAJOR COMMENT

In their introduction, the authors mention the extensive literature on graph structures that promote the evolution of cooperation. I was therefore surprised to see no discussion of how the graph structure introduced in this paper relates to those elsewhere in the literature. What properties does this topology share in common with others that promote cooperation? Do the authors believe these common features to be necessary for a graph to promote the evolution of cooperation?

MINOR COMMENTS

- In the learning model employed by the authors, a focal node changes its strategy to that of the highest-payoff neighboring node, if the payoff of that node is greater than the current payoff to the focal node. In the literature, two relevant payoff specifications have been considered: (i) a node's average/expected payoff (averaged over all possible interactions with neighbors), and (ii) a node's total payoff (so that nodes with more neighbors have the opportunity of a higher payoff). The authors employ specification (ii), which should be explicitly stated in the model setup, since the alternative specification of average payoffs is also reasonable (and common). Perhaps the authors could also discuss the importance of this specification for their results?

- In Figure 5, the 100-x line for the star is exact, and need not be plotted for simulated data. From a state of all cooperation on the star graph, if some nodes are subsequently changed to defection, then cooperation will eventually be restored if and only if the central node was not changed to a defector. If x percent of nodes are changed to defection, then the percentage chance that the central node was not among them is 100-x.

- Especially in the Appendix, it would be useful to give the labels for the nodes in the figures, so that one doesn't have to continually refer to Figure 2 in the main text to remind oneself which are the z , z' , etc. nodes.

===PREPARING YOUR MANUSCRIPT===

===PREPARING YOUR REVISION IN SCHOLARONE===

<https://royalsociety.org/journals/authors/author-guidelines/#supplementary-material> to include a suitable title and informative caption. An example of appropriate titling and captioning may be found at https://figshare.com/articles/Table_S2_from_Is_there_a_trade-off_between_peak_performance_and_performance_breadth_across_temperatures_for_aerobic_sc_ope_in_teleost_fishes_/3843624.

Author's Response to Decision Letter for (RSOS-201958.R0)

See Appendix A.

Decision letter (RSOS-201958.R1)

Dear Dr O'Riordan

On behalf of the Editors, we are pleased to inform you that your Manuscript RSOS-201958.R1 "Introducing a graph topology for robust cooperation" has been accepted for publication in Royal Society Open Science subject to minor revision in accordance with the referees' reports.

We ask that you provide not only the files required below but also that you archive your GitHub deposition in the Zenodo repository (please see guidance at <https://guides.github.com/activities/citable-code/>). Depositing your code ensures it is citeable and more easily discovered. Once you have archived the data in Zenodo, please ensure you include the URL and DOI to the Zenodo deposition in your data access statement, as well as including a citation in your bibliography.

Please submit your revised manuscript and required files (see below) no later than 7 days from today's (ie 23-Feb-2021) date. Note: the ScholarOne system will 'lock' if submission of the revision is attempted 7 or more days after the deadline. If you do not think you will be able to meet this deadline please contact the editorial office immediately.

on behalf of Dr Derek Abbott (Associate Editor) and Marta Kwiatkowska (Subject Editor)
openscience@royalsociety.org

===PREPARING YOUR REVISION===

- If you are providing image files for potential cover images, please upload these at this step, and inform the editorial office you have done so. You must hold the copyright to any image provided.
- A copy of your point-by-point response to referees and Editors. This will expedite the preparation of your proof.

- Ensure that your data access statement meets the requirements at <https://royalsociety.org/journals/authors/author-guidelines/#data>. You should ensure that you cite the dataset in your reference list. If you have deposited data etc in the Dryad repository, please only include the 'For publication' link at this stage. You should remove the 'For review' link.
- If you are requesting an article processing charge waiver, you must select the relevant waiver option (if requesting a discretionary waiver, the form should have been uploaded at Step 3 'File upload' above).
- If you have uploaded ESM files, please ensure you follow the guidance at <https://royalsociety.org/journals/authors/author-guidelines/#supplementary-material> to include a suitable title and informative caption. An example of appropriate titling and captioning may be found at https://figshare.com/articles/Table_S2_from_Is_there_a_trade-off_between_peak_performance_and_performance_breadth_across_temperatures_for_aerobic_scope_in_teleost_fishes_/3843624.

Author's Response to Decision Letter for (RSOS-201958.R1)

See Appendix B.

Decision letter (RSOS-201958.R2)

Dear Dr O'Riordan,

It is a pleasure to accept your manuscript entitled "Introducing a graph topology for robust cooperation" in its current form for publication in Royal Society Open Science.

on behalf of Dr Derek Abbott (Associate Editor) and Marta Kwiatkowska (Subject Editor)
openscience@royalsociety.org

Appendix A

Dear Editor & Reviewers,

We would like to sincerely thank the editor and reviewers for the thoughtful, thorough and constructive comments and also for the time and effort spent reviewing the work and writing the reviews. We are very grateful for the detailed comments and for the opportunity to revise and resubmit our contribution.

We present a point-by-point summary of how we addressed the issues raised.

Several points were made by more than one reviewer; we include the responses for those comments to each of the reviewers

Comments by Reviewer 1

For this model the authors describe some simple (and well-known) phenomena supporting the dominance of cooperation in the final stationary state. I cannot recommend the publication of this work because of the absence of sufficiently important and new results.

We thank the reviewer for bringing our attention to the issues that the novelty of the work was not clearly articulated in the first original draft. We added some text to highlight where the work differs from existing work. We also attempt to place the work in the context of existing work. In particular we note how this topology works for the prisoner's dilemma. It isn't limited to promoting a mutant which achieves a higher payoff than others. The topology also supports cooperation in scenarios where there are many perturbations whereas much of the existing similar research adopts a simpler interaction model.

The most relevant topological features (irregularity in the number of neighbours, overlapping small cliques/triangles or comets versus stars) - that support the spreading of cooperation - are well investigated in the literature and surveyed in several reviews. Unfortunately, the authors don't use this knowledge in the interpretation of their results.

We welcome the comment that we didn't use existing known structures in our interpretation. We attempted to address this limitation by providing some further comparisons to topological features such as comets and stars. A comet in our interaction model and update mechanism: we have identified that a complete graph is a Resilient graph for defection. Depending on the size of the star subgraph, the cooperation robustness of the comet will be somewhere between that of a complete graph (0% cooperation if 1 or more defectors) up to that of a star (cooperation is highly dependent on whether or not the central node is perturbed) which means that the performance of a comet is less than that of a star. The only similarity between a comet and our graph is the presence of a star subgraph (or more in our structure).

One could argue that due to the unique nature of the complete graph (shortest longest path of 1) and the star graph (extreme degree distribution), they can be useful tools/components in the design

of structures that behave in specific ways but the actual effects of these structures is highly dependent on the interaction model and update mechanism used, with the structures possibly being irrelevant in some scenarios.

In much of the previous work, only one perturbation is considered (presumably because it is hard or unlikely to find a graph with a very strong effect for their particular interaction model and update mechanism).

The chain-like topological structure of connections and the applied dynamical rule are not realistic. I think that the scientific value of this work can be improved by considering the effect of stochastic imitations, too. The one-dimensional feature of this connectivity structures can be exploited by determining the average velocity of the boundary (separating homogeneous regions of cooperation and defection) for different parameters.

In the paper we present the topology in its most basic form. We include this basic form for clarity purposes. There are many extensions and variations. We have included the following text to the paper to indicate what modifications could be made to the topology while maintaining its core functionality:

“Given the definition of our graph, the number and ratios of minor nodes can be varied while maintaining the core functionality of the graph; we have identified a series of additional methods we can use to grow or extend our graph while maintaining the core functionality. These include:

- 1. We can add additional enabler nodes: this may alter the degree distribution of our graph potentially allowing for other payoffs and could make cooperation spread faster in some cases.*
- 2. We can add edges between minor nodes both connected to the same large node or to different large nodes. This may alter the degree distribution of our structure. It could slow the spread of cooperation. There is a limit to which we can do this before losing the core functionality.*
- 3. A critical node can be connected to multiple large non-critical nodes (B) (this will need the addition of enabler nodes).*
- 4. A large non-critical node can be connected to other large non-critical nodes (this will need the addition of enabler nodes).*
- 5. A large non-critical node can be connected to multiple critical nodes (this will need the addition of enabler nodes).*

Given points 3 and 5 above, if we consider a critical node subgraph to be a "node" and the large non-critical node subgraph connecting two critical node subgraphs to be the "edge" between two "nodes", then one can create any undirected "graph" mimicking some of the properties of those actual graph. Examples include fractal graphs and scale-free graphs. The many ways to grow our topology and their effects on the graph properties and the limit to which we can maintain core functionality will be explored in future work.”

We agree that the one dimensional nature of the topology could be exploited; however several of the means of extending the graph does not have that same property.

With regard to introducing an imitation model that is stochastic in nature, we agree that this would be an interesting avenue to explore which we plan to do in the future. There are many sources of potential stochasticity in the model. We discuss some in the paper under the broader category of noise.

The descriptions of the model and method are verbose. A more concise survey of these details would improve the presentation.

We have tried to improve the presentation by including more detailed descriptions of certain aspects such as the calculation of the overall payoff.

Reviewer 2:

After reading the manuscript, I have found it interesting, but I have some following questions or comments on this work.

Thank you for your encouraging words; we are glad you found it interesting.

In this work, the authors use a simple strategy update mechanism where agents copy the best performing strategy of their neighborhood. I think that the graph topology for robust cooperation the author introduced is sensitive to this strategy update mechanism. Is that true? I wonder whether the graph topology still has the property when other strategy update mechanisms are considered.

The graph topology presented is specific to this particular interaction model and update mechanism. While it is possible for the topology to have similar functionality with other interaction models and update mechanisms, we expect that in general it will not support cooperation in the same manner. However, if there is a topology which offers similar functionality, in those specific cases, it might be a modified version of our topology, or it might share some elements in common.

We have included several paragraphs which discuss how our topology might perform under different interaction models or update mechanisms.

In this work, the authors consider the synchronous updates under which all the players calculate their payoffs and potentially update their strategy at the same time. I wonder whether their conclusion is still valid when the asynchronous updates are considered.

We consider the asynchronous update to be a different update mechanism and so the previous point applies. We have included the following paragraphs to discuss the effects of asynchronous updates:

“In the update mechanism presented in this paper all nodes update their strategy at the same time (synchronous update). With a different update mechanism which uses asynchronous updates, where only one node updates at a time, the behaviour of the population might change. We will discuss two types of asynchronous updates:

- *When a node updates, all nodes recalculate their payoff: Our topology is still robust to any one perturbation. Eventually defection spreads enough such that cooperation can spread back. Defection still can't spread from node B to node A or from node a to node A or from node z to either node A or B. There may be some changes to the overall robustness of the*

graph, and the graph will take many more turns to stabilise (either to full cooperation or defection). The star topology and the complete graph have the same robustness, but again it will take multiple turns for the population to stabilise.

- *A node recalculates its payoff only when it updates (once initially before comparing to others and once the update finishes): Our topology is still robust to any one perturbation. As in the previous case, defection eventually spreads enough so that cooperation can spread back. As before, defection still can't spread from node B to node A or from node a to node A or from node z to either node A or B. A defective critical node should maintain a higher payoff for longer (we have to wait for its minor nodes to defect and then it has to update); eventually cooperation will spread once the defected critical node updates to a lower payoff. Again there might be some changes on the overall robustness of the graph and the graph will take many more turns to stabilise. The star topology will obtain the same robustness. For the complete graph robustness stays the same, it is possible for cooperation to spread (for example after initial perturbation one of cooperators is selected then it will defect and then if all other defectors are selected one at a time then no defector will have the original payoff which when the next defector is selected may allow the cooperators, who have their original payoff, to have a higher payoff allowing cooperation to spread) but at no point can cooperation spread to more than the original number of cooperators which means defection will eventually spread to the whole graph."*

Other possible asynchronous updates can be considered, including ones in which a fraction of all nodes update at the same time.

Reviewer 3:

Is it reasonable to assuming perfect information/observation in terms of observing the best neighbouring payoffs and subsequently copying strategies? What is the impact of noise in these aspects?

We agree that perfect information/observation is not always possible or realistic. The graph topology presented is specific to this particular interaction model and update mechanism. We consider the presence of noise to be a different update mechanism. A different structure might be more suitable for each different kind of noise. We have included the following paragraphs to discuss the effects of several forms of noise:

"In the current model, the players have perfect knowledge and behaviour. In an update mechanism that contains noise, this is no longer the case. We discuss two types of noise:

- *Noise giving the player a chance to incorrectly read the payoff obtained by a neighbouring node when it decides how to update: We can no longer guarantee that our structure is fully robust to any one perturbation (for example if B defects, both A and A' might read its payoff as being larger than theirs which would allow the defection to spread to them which eventually would probably make the defection to spread to all nodes). The overall robustness*

of the graph would also be affected the extent of which is hard to say since the noise will help with the spread of both defection and cooperation. To reduce the effects of the noise on our structure we could have more minor nodes and enabler nodes. The change in payoff caused by noise will need to be very high to affect the robustness of the star topology (and even higher for larger stars). In a complete graph it will now be possible for a defector to read a cooperator's payoff as being higher, which will spread cooperation to the defector, at the same time defection will probably spread to the cooperator (and most of all the other cooperators), so on average the robustness of the complete graph should stay the same.

- *Noise giving the player a chance to incorrectly read the strategy of a node (with no change to its payoff): We can no longer guarantee that our structure is fully robust to any one perturbation. In a fully cooperating critical node subgraph, the critical node still can't be turned to defection; the B nodes will read A nodes as defectors from time to time causing them to defect, minor nodes will turn to the opposite strategy of the critical nodes due to noise which makes the critical node weaker when it cooperates possibly allowing the B node to influence it, it also makes defector critical nodes stronger. It is hard to predict how much this noise will reduce the overall robustness of our structure, but is expected that at all times a fraction of the minor nodes will always be defective due to wrongly perceiving the strategy of their cooperating large nodes. To combat the effect of this noise, we can increase the number of a and b nodes such that the ratio a/b becomes larger making it more difficult for node B to influence node A. On a star graph the overall robustness of the graph will still mostly depend on whether or not the central node is the initial perturbation, but the robustness will further be reduced since at all times a fraction of the one-degree nodes will defect due to the noise. On the complete graph a fraction of the nodes will cooperate at all times due to noise, but this is reduced since the perceived cooperator will have the same payoff as the defectors so the selected strategy is randomly selected from one of the defectors or the defectors incorrectly read as cooperators, increasing the overall robustness for cooperation."*

Since each node plays a game with each of its neighbours the number of edges determines the available payoff. Thus, the "hub" nodes will clearly be more influential, not just because more neighbours can potentially copy them, but because in a broadly cooperative environment it is the hubs that received the high (cumulative) payoffs in a given round. What would be the impact of copying based on "average interaction payoff", which would better reflect the relative effectiveness of the strategies?

It is correct that that the available payoff is dependent on the degree of a node. We had previously shortly discussed this in Section 6 and we have added the following paragraph to cover the issue in more detail.

"Given our current interaction model and update mechanism, the payoff of a node is highly dependent on the degree of that node. Generally as a node's degree increases so does their payoff. It would be interesting to explore an alternative payoff mechanism where we normalize the payoff, by dividing it by the node's degree, resulting in an average interaction payoff. Given this approach, the topology presented in this paper will no longer function in the same manner. For example, if a minor node defects it will always have a payoff of T when the large node it is connected to cooperates. The maximum payoff of this large node is R, which means cooperation can't spread to the perturbed

*minor node and in fact it is guaranteed to spread defection. This implies that star graphs or structures containing them are not suitable for maintaining cooperation for this particular interaction model and update mechanism. In the complete graph if we have m defectors and n cooperators the payoff of defectors is $(n * T + m * P - P)/(n + m)$ and that of cooperators is $(n * R + m * S - R)/(n + m)$ given $T > R > P > S$ then the defectors will always have a higher payoff than cooperators.”*

The comparison against the star topology is informative, but it would also be useful to consider some more realistic topologies.

We agree that comparing to other topologies would be an interesting avenue. We have included the following paragraph discussing some of the difficulties in comparing our topology with others and outlining some arguments as to why ours should outperform many topologies once we reach sufficiently large graphs:

“One of the reasons it is hard to compare the robustness of two graphs is that this robustness is not only dependent on the interaction model and update mechanism but also on the actual payoffs used. As we have discussed, our structure only functions properly for a certain range of payoffs, whereas, for example, the robustness of a $k=4$ lattice graph (a graph in which each node is placed on a grid and it is connected to its 4 closest neighbours with the nodes on the edges being neighbours with the nodes on the opposite edge), will vary considerably across multiple ranges of payoffs (the behaviour will be different when a defector connected to two cooperators and two defectors has either a higher payoff than a cooperator connected to 3 cooperators and a defector compared to the case in which it is lower). In general, for sufficiently large graphs our topology should outperform others, for the payoffs in which it functions properly, given the fact that its robustness improves as it grows while that of others should in general stay unchanged.”

The method of extended the graph by connecting in a "line topology" seems overly restrictive. Have other extensions been considered? Given the discussion on the characteristics of a star topology it seems that this form of extension is also possible.

We agree that there may be many other ways of extending the graph and the line topology is just one of many. We have considered other ways to extend the topology which we would cover in another paper, we have added the following paragraphs to discuss these extensions in more detail:

“Given the definition of our graph, the number and ratios of minor nodes can be varied while maintaining the core functionality of the graph; we have identified a series of additional methods we can use to grow or extend our graph while maintaining the core functionality. These include:

- 1. We can add additional enabler nodes: this may alter the degree distribution of our graph potentially allowing for other payoffs and could make cooperation spread faster in some cases.*

2. We can add edges between minor nodes both connected to the same large node or to different large nodes. This may alter the degree distribution of our structure. It could slow the spread of cooperation. There is a limit to which we can do this before losing the core functionality.
3. A critical node can be connected to multiple large non-critical nodes (B) (this will need the addition of enabler nodes).
4. A large non-critical node can be connected to other large non-critical nodes (this will need the addition of enabler nodes).
5. A large non-critical node can be connected to multiple critical nodes (this will need the addition of enabler nodes).

Given points 3 and 5 above, if we consider a critical node subgraph to be a "node" and the large non-critical node subgraph connecting two critical node subgraphs to be the "edge" between two "nodes", then one can create any undirected "graph" mimicking some of the properties of those actual graph. Examples include fractal graphs and scale-free graphs. The many ways to grow our topology and their effects on the graph properties and the limit to which we can maintain core functionality will be explored in future work."

When considering the introduction of defectors, what is the impact of more targeted perturbations (rather than random). For example, in the context of influence maximisation (which although rather different has parallels in terms of "strategies" potentially spreading in a connected topology), it is common to use simple targeting such as degree or degree-discount. What would be the impact of such interventions here? For example, is cooperation still achieved if the A nodes or B nodes are targeted? Graphs such as that in Figure 2, would be robust to 2 "a" nodes being perturbed, but the situation of perturbing A and A' is not clearly discussed. Some further analysis of the cases that are not cooperative in Figure 5 would be informative here.

In the work, we have considered random perturbation. The suggestion of targeted perturbations is an interesting one and deserving of further attention. A more targeted initial perturbation will lower the robustness of the graph. In the case of 2 perturbations, in Figure 2, in which both A and A' are perturbed then defection will spread to the whole graph since at no time there will be any critical node subgraph fully cooperating. We added the following paragraph to discuss the effects of a more targeted initial perturbation:

"In our experiments, the initial perturbations are at random. If instead the perturbations were more targeted with higher degree nodes selected more often, then it will lower the robustness of our structure. The exact effects would depend on the strength/precision of targeting; for example, if half the critical nodes are guaranteed to be targeted while all other perturbations are at random, then the robustness of the graph would be very similar to that of random targeting in a graph of half the size. If critical nodes are 20 times more likely to be targeted compared to other nodes, then the robustness of the graph would be very similar to that of random targeting in a graph 20 times smaller. As discussed in the definition of critical nodes, if at any time a critical node and all its minor nodes cooperate, then cooperation will spread from it. Regarding our structure, as seen in Figure 2, we showed that it is robust to any 1 perturbation; when considering 2 perturbations the graph recovers to full cooperation as long as 1 of the critical nodes is not perturbed (given the definition of critical nodes and the fact that node B would have a lower payoff than the defected critical nodes).

To reduce the effects of a more targeted initial perturbations, our structure would be constructed such that the large nodes would have the minimum number of minor nodes reducing the chance that the large node is targeted, we can also add additional edges between minor nodes making them more likely to be targeted. This targeted perturbation will similarly reduce the robustness of the star graph, while for the complete graph it will have no effect since all nodes in it have the same degree."

For non cooperative cases in Figure 5, we have examined several examples and for all the cases, we found that at no time was any one of the critical node subgraphs fully cooperating. This is what we expected given the definition of critical nodes.

How does the proposed graph structure relate to structures observed in more general synthetic graphs (e.g. random, small-world, scale-free etc.) and real-world graphs (e.g. those available from the Stanford Network Analysis Project)? Are subgraphs, or motifs, of the proposed form observed elsewhere? Does their introduction add robustness to existing graphs?

We added the following paragraph to discuss the presence or introduction of our topology in other graphs:

"When looking at sufficiently large random, scale-free or real-world graphs, we would generally find star like subgraphs in them. Given this, it would be possible to slightly modify graphs such that our graph is present as a subgraph, which would improve the robustness of the whole graph due to the robustness of our subgraph."

The proposed structure is related to other graph structures and motifs which have similar effects, and this needs further discussion in the related work section. For example, in the norm emergence literature there is a notion of "self reinforcing" structure, which has a similar effect to the proposed graph structure

Thank you for the references and the suggestions.

We have included the following paragraph to discuss the similar effect found in our topology and in "self reinforcing" structures:

"Subgraphs of our topology, critical node subgraphs, have similar functionality to that of "self reinforcing" structures [24], they both stop other strategies/conventions from outside the subgraph to spread into the subgraph, but our subgraph differs in the sense that they are also responsible of spreading the cooperation to nodes outside the subgraph, nodes which in their turn spread cooperation further due to the subgraph they belong to, we present proof of this property."

The phrasing "we define the following category of graphs, 'resilient graphs'" is a little misleading, since there is no formal definition and the only examples discussed are the proposed structure and a complete graph. If this really is a new category of graph more discussion is needed.

We agree; we provided no formal definition. We have updated that paragraph to reflect that:

“We introduce the following category of graphs, “Resilient Graphs”, a resilient graph is a graph which will support one strategy while hindering all others. Resilient graphs are a more generalised version of suppressors, which make mutants less likely to spread to the whole population. For Resilient Graphs there is no need to know if a strategy/mutation is advantageous which allows us to look at more complex games. The graph topology presented in this paper is a resilient graph for cooperators participating in the Prisoner’s Dilemma with a strategy update mechanism where the best performing player in the neighbourhood is copied.”

Section 5 refers to “our robustness measure”, but I don't see a clear definition of what this is.

We replaced “our robustness measure” with “our robustness measurement, the average percentage of cooperators at the end of simulation given a certain percentage of initial defectors”.

I suggest rephrasing the future work to be more general, rather than identifying areas such as “robustness of $k = 4$ n by n lattice graphs”.

We rephrased that sentence to: *“We will explore the robustness of standard graphs when using the same interaction model and update mechanism as in this paper.”*

Finally, the balance between the main body of the paper and the appendices might be improved by moving some of the technical content from the appendices into the main body.

We had considered the balance and were aware that much of technical content had been moved to appendices. Our reasoning was that the derivation of the proof would be of interest to fewer readers. However, we have attempted to move some of the content to the main body.

We have introduced at the end of section 2 the list of requirements derived in Appendix A and B, we do not believe the exact calculations of how were derived are of interest to most readers.

In section 5 we have changed *“The simulations ended up finding a solution to our graph as long as $T/R < R/P$.”* to *“The simulations ended up finding a solution to our graph as long as $T/R < R/P$ (this requirement has also been obtained in Appendix C).”*

The results from Appendix D have already been presented in the main paper.

In response to comments and suggestions by a number of reviewers, we have added content in a number of sections. We believe this will also help the balance of the paper.

Reviewer 4:

In their introduction, the authors mention the extensive literature on graph structures that promote the evolution of cooperation. I was therefore surprised to see no discussion of how the graph structure introduced in this paper relates to those elsewhere in the literature. What properties does this topology share in common with others that promote cooperation? Do the authors believe these common features to be necessary for a graph to promote the evolution of cooperation?

We have introduced the following paragraphs to compare our topology with others with similar effect:

“Whilst the topology described in this paper has some key similarities with existing work - the presence of star-like graphs which have been investigated on their own and which are also a component of comet graph structures [23].

The effect of the topology on the outcome is dependent on the interaction model; much of the recent work adopt an abstract interaction model where mutants who outperform the initial population are considered. In our work, we consider a more complex interaction model (the prisoner’s dilemma) where the payoff received by an agent is dependent on its neighbours. The increased complexity of this interactions model requires a more complex topology to help maintain robust cooperation.”

And

“Subgraphs of our topology, critical node subgraphs, have similar functionality to that of “self reinforcing” structures [24], they both stop other strategies/conventions from outside the subgraph to spread into the subgraph, but our subgraph differs in the sense that they are also responsible of spreading the cooperation to nodes outside the subgraph, nodes which in their turn spread cooperation further due to the subgraph they belong to, we present proof of this property.”

Regarding the universal features necessary for cooperation, to answer that we would need to look at many different interaction models and update mechanisms before we can say, which is beyond the scope of this paper. We have added the following paragraph in section 6 stating that we will explore these features in future work:

“Once Resilient structures have been explored in many interaction models and update mechanisms we can take a look at what common characteristics this structures have and see if some characteristics are only viable for a subsection of interaction models and update mechanisms.”

In the learning model employed by the authors, a focal node changes its strategy to that of the highest-payoff neighboring node, if the payoff of that node is greater than the current payoff to the focal node. In the literature, two relevant payoff specifications have been considered: (i) a node's average/expected payoff (averaged over all possible interactions with neighbors), and (ii) a node's total payoff (so that nodes with more neighbors have the opportunity of a higher payoff). The authors employ specification (ii), which should be explicitly stated in the model setup, since

the alternative specification of average payoffs is also reasonable (and common). Perhaps the authors could also discuss the importance of this specification for their results?

We do use specification (ii) we have added the following text to make it clearer:

" A player's payoff is calculated as the sum of all payoffs obtained by the player when interacting with their neighbours in the current turn."

We have previously shortly discussed this average/normalized payoff in Section 6, we have added the following paragraph to discuss it in more detail:

*"Given our current interaction model and update mechanism, the payoff of a node is highly dependent on the degree of that node. Generally as a node's degree increases so does their payoff. It would be interesting to explore an alternative payoff mechanism where we normalize the payoff, by dividing it by the node's degree, resulting in an average interaction payoff. Given this approach, the topology presented in this paper will no longer function in the same manner. For example, if a minor node defects it will always have a payoff of T when the large node it is connected to cooperates. The maximum payoff of this large node is R, which means cooperation can't spread to the perturbed minor node and in fact it is guaranteed to spread defection. This implies that star graphs or structures containing them are not suitable for maintaining cooperation for this particular interaction model and update mechanism. In the complete graph if we have m defectors and n cooperators the payoff of defectors is $(n * T + m * P - P)/(n + m)$ and that of cooperators is $(n * R + m * S - R)/(n + m)$ given $T > R > P > S$ then the defectors will always have a higher payoff than cooperators."*

In Figure 5, the 100-x line for the star is exact, and need not be plotted for simulated data. From a state of all cooperation on the star graph, if some nodes are subsequently changed to defection, then cooperation will eventually be restored if and only if the central node was not changed to a defector. If x percent of nodes are changed to defection, then the percentage chance that the central node was not among them is 100-x.

The results for the star graph are mostly deterministic (which we stated in the paper), but it is not always true that it is dependent on only the chance of the hub node to defect (for example if the central node cooperates, $S=0$ and all 1 degree node defect then the hub node will have a payoff of 0 while the 1 degree nodes will have a payoff of T making the hub to defect, there are other possible examples such as those with low S and R values and with high T values but they are affected by the size of the star graph).

Especially in the Appendix, it would be useful to give the labels for the nodes in the figures, so that one doesn't have to continually refer to Figure 2 in the main text to remind oneself which are the z, z', etc. nodes.

We have updated the figure 3 and figures 6 to 17 to include labels.

Yours sincerely,

Alex Locodi & Colm O'Riordan

Appendix B

Dear editors and reviewers,

We would like to take this opportunity to thank the reviewers for their helpful and insightful comments on the paper and for the time and effort spend in reviewing our paper.

We believe the comments and suggestions have resulted in stronger paper and contribution.

We have archived your GitHub deposition in the Zenodo repository and have included a citation to said repository.

Yours sincerely,

Alex Locodi & Colm O'Riordan